# Constructing and optimizing 3D atlases from 2D data with application to the developing mouse brain

David M Young[1,2], Siavash Fazel Darbandi[1], Grace Schwartz[1], Zachary Bonzell[1], Deniz Yuruk[1], Mai Nojima[1], Laurent C Gole[2], John LR Rubenstein[1], Weimiao Yu[2], Stephan J Sanders[1]*

[1]Department of Psychiatry and Behavioral Sciences, UCSF Weill Institute for Neurosciences, University of California, San Francisco, San Francisco, United States; [2]Institute of Molecular and Cell Biology, Agency for Science, Technology and Research, Singapore, Singapore

**Abstract** 3D imaging data necessitate 3D reference atlases for accurate quantitative interpretation. Existing computational methods to generate 3D atlases from 2D-derived atlases result in extensive artifacts, while manual curation approaches are labor-intensive. We present a computational approach for 3D atlas construction that substantially reduces artifacts by identifying anatomical boundaries in the underlying imaging data and using these to guide 3D transformation. Anatomical boundaries also allow extension of atlases to complete edge regions. Applying these methods to the eight developmental stages in the Allen Developing Mouse Brain Atlas (ADMBA) led to more comprehensive and accurate atlases. We generated imaging data from 15 whole mouse brains to validate atlas performance and observed qualitative and quantitative improvement (37% greater alignment between atlas and anatomical boundaries). We provide the pipeline as the MagellanMapper software and the eight 3D reconstructed ADMBA atlases. These resources facilitate whole-organ quantitative analysis between samples and across development.

*For correspondence:
stephan.sanders@ucsf.edu

Competing interests: The authors declare that no competing interests exist.

## Introduction

Anatomical atlases have played a crucial role in research into organogenesis, anatomy, physiology, and pathology (*Marquart et al., 2017*; *Chon et al., 2019*; *Liu et al., 2020*) and have been used for a wide range of applications, including investigating developmental biology (*Kunst et al., 2019*), defining stereotactic coordinates (*Franklin and Paxinos, 2013*), identifying patterns of neural activity associated with behavior (*Renier et al., 2016*; *Kim et al., 2017*), and integrating morphological, transcriptomic, and neural activity data across species (*Marquart et al., 2017*). Ongoing initiatives continue to develop these resources, including the Human Cell Atlas (*Regev et al., 2019*), the BRAIN Initiative (*Ecker et al., 2017*; *Koroshetz et al., 2018*), the Human Brain Project (*Amunts et al., 2016*; *Bjerke et al., 2018*), and centralized repositories of data, including the Allen Brain Atlas (*Lein et al., 2007*; *Jones et al., 2009*; *Ng et al., 2007*).

Macroscale imaging, such as magnetic resonance imaging (MRI), routinely generates data from intact whole organs (*Huang and Luo, 2015*; *Thompson et al., 2014a*; *Jahanshad et al., 2013*; *Sudlow et al., 2015*; *Miller et al., 2016*; *Alfaro-Almagro et al., 2018*; *Walhovd et al., 2018*) in contrast to microscale imaging, such as light microscopy, which has typically relied on physically cutting thin sections leading to misalignment artifacts. Technological advances in microscopy have facilitated a dramatic increase in imaging of high-resolution, intact whole organs using serial two-photon tomography (STPT) (*Ragan et al., 2012*) or tissue clearing techniques (e.g. CLARITY [*Chung et al., 2013*], 3DISCO [*Ertürk et al., 2012*], and CUBIC [*Susaki et al., 2014*]) and lightsheet microscopy

**eLife digest** The research community needs precise, reliable 3D atlases of organs to pinpoint where biological structures and processes are located. For instance, these maps are essential to understand where specific genes are turned on or off, or the spatial organization of various groups of cells over time.

For centuries, atlases have been built by thinly 'slicing up' an organ, and then precisely representing each 2D layer. Yet this approach is imperfect: each layer may be accurate on its own, but inevitable mismatches appear between the slices when viewed in 3D or from another angle.

Advances in microscopy now allow entire organs to be imaged in 3D. Comparing these images with atlases could help to detect subtle differences that indicate or underlie disease. However, this is only possible if 3D maps are accurate and do not feature mismatches between layers. To create an atlas without such artifacts, one approach consists in starting from scratch and manually redrawing the maps in 3D, a labor-intensive method that discards a large body of well-established atlases.

Instead, Young et al. set out to create an automated method which could help to refine existing 'layer-based' atlases, releasing software that anyone can use to improve current maps. The package was created by harnessing eight atlases in the Allen Developing Mouse Brain Atlas, and then using the underlying anatomical images to resolve discrepancies between layers or fill out any missing areas. Known as MagellanMapper, the software was extensively tested to demonstrate the accuracy of the maps it creates, including comparison to whole-brain imaging data from 15 mouse brains. Armed with this new software, researchers can improve the accuracy of their atlases, helping them to understand the structure of organs at the level of the cell and giving them insight into a broad range of human disorders.

(*Chhetri et al., 2015*; *Meijering et al., 2016*; *Mano et al., 2018*; *Hillman et al., 2019*; *Cai et al., 2019*; *Ueda et al., 2020*). These 3D microscopy methods have been applied to map cytoarchitecture and cellular connectivity in complex tissues, including the brain (*Cai et al., 2019*; *Kim et al., 2015*; *Murakami et al., 2018*), heart (*Gilbert et al., 2019*; *Bai et al., 2015*), and lung (*Li et al., 2012*; *Ardini-Poleske et al., 2017*).

Accurate analysis of imaging data from whole organs requires atlases of the corresponding organ, species, developmental age, resolution, and dimensions (*Allen Institute for Brain Science, 2017*; *Susaki and Ueda, 2016*; *Niedworok et al., 2016*; *Watson and Watkins, 2019*). The progression to 3D microscopy data necessitates 3D anatomical atlases to reference data to existing anatomically-defined labels (*Puelles and Rubenstein, 2003*; *Watson et al., 2012*). Although numerous atlases based on 2D physical sections exist (*Franklin and Paxinos, 2013*; *Ardini-Poleske et al., 2017*; *Savolainen et al., 2009*; *de Boer et al., 2012*; *Thompson et al., 2014b*), most suffer from label misalignments or insufficient resolution in the third dimension, limiting their utility for 3D imaging (*Wang et al., 2020*). The most recent version of the Allen Common Coordinate Framework atlas (CCFv3) of the adult (P56) C57BL/6J mouse brain represents one of the most complete 3D atlases at microscopic resolution to date (*Wang et al., 2020*). A team of neuroanatomists and illustrators redrew structures natively in 3D, sometimes using the original annotations as seeds to guide the redrawing process, producing 658 structural annotations (*Wang et al., 2020*). Even with this monumental effort, 23% of labels from the original Allen Reference Atlas were not included in the 3D version, largely due to the time and labor constraints of further manual curation (*Wang et al., 2020*). To date, this titanic effort to generate a 3D atlas at microscopic resolution has not been repeated for most combinations of organ, species, and developmental stage, despite the numerous existing 2D-based atlases.

Automated methods to convert existing 2D-derived atlases into fully 3D atlases could leverage existing, established atlases without the time-consuming effort required for manual 3D curation, or provide the substrate for more rapid manual fine-tuning. This conversion needs to address multiple limitations in atlases based on 2D physical sections, as demonstrated by the challenges of using the existing 3D reconstruction of eight mouse brains, each at a different developmental time point, in the Allen Developing Mouse Brain Atlas (ADMBA) (*Thompson et al., 2014b*; *Figure 1A*). First, all atlases in the ADMBA are missing labels across large regions, typically the lateral planes of one

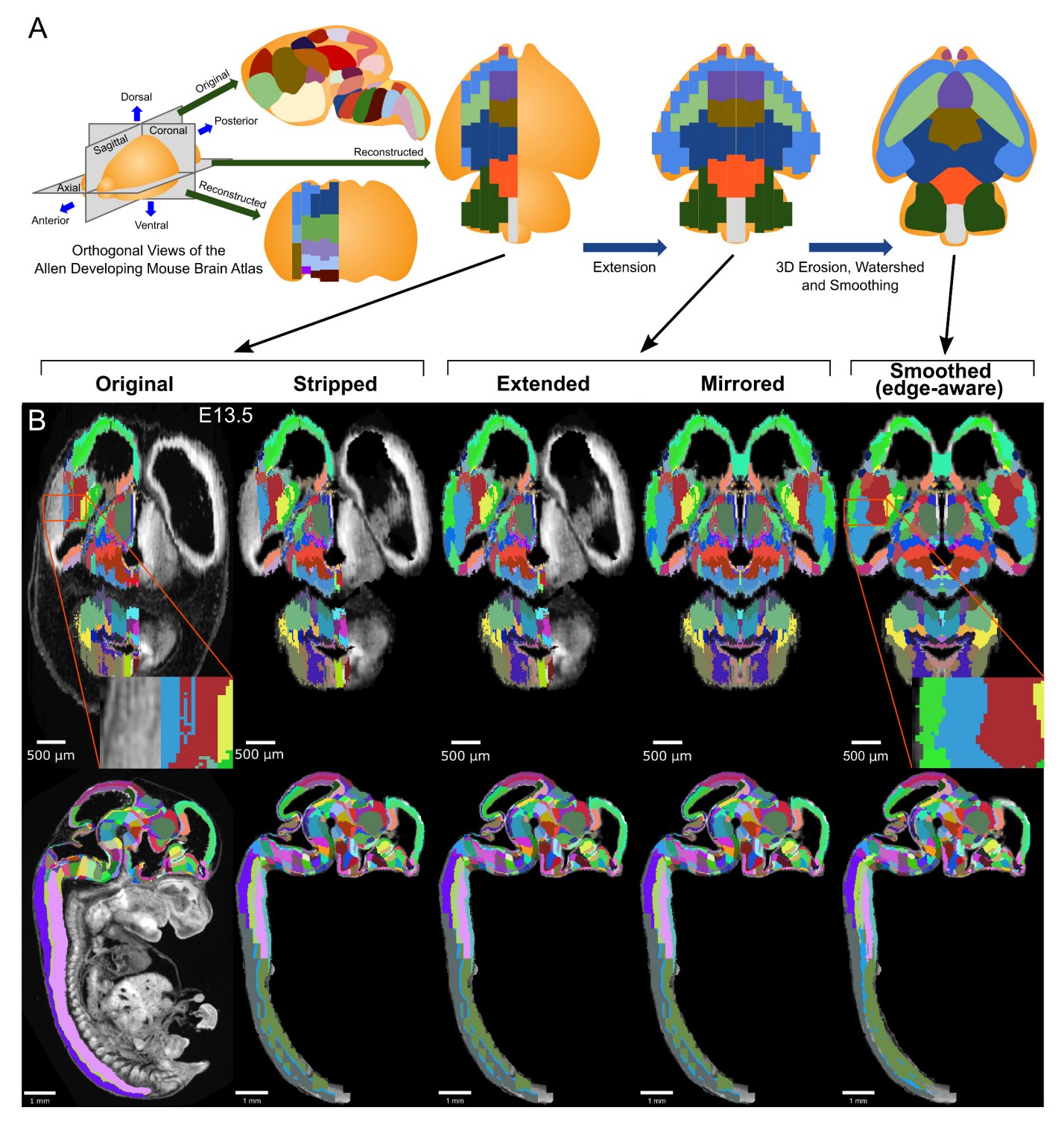

**Figure 1.** 3D atlas refinement pipeline overview and full example in the E13.5 atlas. (**A**) 2D-derived atlases, such as those in the Allen Developing Mouse Brain Atlas, are smooth and consistent in the sagittal plane in which they were annotated. However, in the 3D reconstructions of these 2D sagittal planes, the coronal and axial planes reveal missing sections and jagged edges. To improve their performance for annotating 3D data, the lateral edges are extended to complete the labeled hemisphere. A 3D rotation is applied to bring the brain parallel to the image borders, then both the completed hemisphere labels and underlying microscopy sections are mirrored across the sagittal midline to complete coverage in the opposite hemisphere. To improve anatomical registration, the labels are each eroded and re-grown through a 3D watershed, guided by the anatomical edge map. To smooth the final product, labels are iteratively filtered with a morphological opening operation, or a closing operation for small, disconnected

*Figure 1 continued on next page*

*Figure 1 continued*
labels that would otherwise be lost. (B) The pipeline illustrated in axial (top) and sagittal (bottom) views on the ADMBA E13.5 atlas, which requires the full pipeline shown in 'A', including an additional step to strip out non-CNS tissue from the original whole-embryo imaging. The nomenclature for pipeline steps shown here is used consistently throughout the manuscript. Insets of the 'Original' and 'Smoothed (edge-aware)' lateral regions highlight the label extension and smoothing. A spreadsheet mapping colors to label names and IDs in the Allen ontology for each atlas in this manuscript can be found in *Supplementary file 1*.

hemisphere and the entire opposite hemisphere. Second, warping and asymmetrical placement of samples in the histological images prevents simply mirroring labels from the annotated to the non-annotated hemisphere. Third, as noted in the original Allen Reference Atlas (*Ng et al., 2007*; *Niedworok et al., 2016*; *Wang et al., 2020*), the assembled 3D volume is smooth in the sagittal planes, in which the atlas annotations were drawn, but suffers from substantial edge artifacts in the coronal and axial planes (*Figure 1A*).

Several automated approaches have partially addressed some of these issues. The Waxholm Space rat brain atlas used two standard reference atlases as templates to yield an atlas with relatively smooth 3D structures through manual and semi-automated segmentation, although some label edge artifacts remain (*Papp et al., 2014*). *Erö et al., 2018* performed non-rigid alignments between histological slice planes of the Common Coordinate Framework (CCFv2, 2011), applying the same deformations to the label images. This registration improved alignment; however, the labels often appear to extend beyond the underlying histological planes. Rather than registering planes, *Niedworok et al., 2016* applied a smoothing filter on the annotations themselves to reduce label artifacts, though many label edge irregularities persisted.

Here, we present a pipeline to automatically generate fully 3D atlases from existing 2D-derived partial atlases and apply it to the full ADMBA. We performed three steps (*Figure 1A*): (1) Extension and mirroring of 2D data to approximate missing labels, (2) Smoothing with 3D watershed and morphological methods to align jagged edges between 2D sections, and (3) Edge detection from microscopy images to guide the 3D watershed to align boundaries with anatomical divisions, which we call 'edge-aware' refinement (example atlas shown in *Figure 1B*). We demonstrate qualitative and quantitative improvements over the existing 3D ADMBA. To assess the resulting 3D atlas, we generated 3D imaging data from 15 C57BL/6J wild-type mouse brains at postnatal day 0 (P0) using tissue clearing and lightsheet microscopy. The 3D labels in our refined atlas match the brain structures better than the initial atlas, as demonstrated by closer alignment between labels and anatomical edges and decreased variability within labels at the cellular level. We provide the pipeline as open source software called MagellanMapper to apply the algorithms to equivalent atlases and provide the resulting ADMBA 3D atlases as a resource for immediate application to automated whole-brain 3D microscopy analysis.

## Results

### Allen developmental mouse brain atlas

The ADMBA contains data for eight mouse brains, each at a different development stage: embryonic day (E)11.5, E13.5, E15.5, E18.5, postnatal day (P) 4, P14, P28, and P56. The data are generated from 158 to 456 serial 2D sagittal sections of 20–25 µm stained with Nissl or Feulgen-HP yellow and imaged with a light microscope at a lateral resolution of 0.99 to 1.049 µm. For the three earliest stages of development, E11.5, E13.5, E15.5, the entire embryo was imaged, rather than just the brain. Each of the eight atlases are annotated with expertly curated labels of brain structures (*Thompson et al., 2014b*; *Allen Institute for Brain Science, 2013b*) in a hierarchy starting with the largest structures (e.g. neural plate) extending through 13 sublevels to the smallest annotated structures (e.g. parvicellular part of Lat). Viewed sagittally, these labels cover the majority of tissue in each brain with smooth, anatomically aligned edges (*Figure 2A*); however, only sections on the left side of each brain are annotated and, for six atlases, labels are not present for the most lateral 14–24% sagittal planes of the brain (4–5% by volume; *Figure 2—figure supplement 1*; *Figure 2B*). Labels extend slightly beyond the midline for several atlases, helping to annotate brains with a midline skew, yielding a median label to atlas volume ratio of 51%.

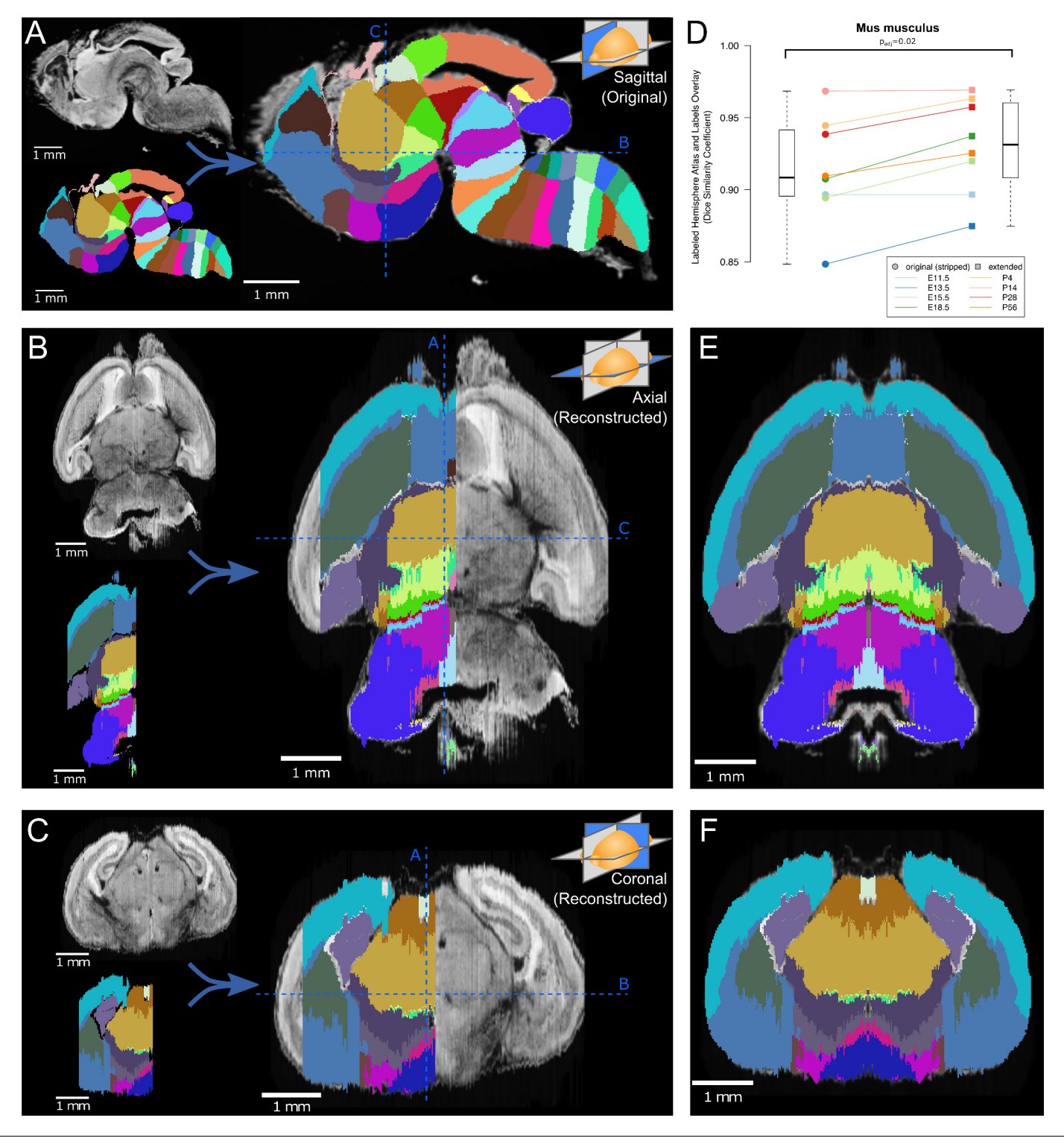

**Figure 2.** Atlas lateral extension and mirroring for complete brain annotation. (**A**) The original E18.5 atlas labels (bottom left), viewed sagittally, demonstrate smooth borders and close correspondence with the underlying microscopy images (upper left; overlaid on right). The dashed blue lines show the sections viewed in 'B' and 'C'. (**B**) When viewed axially, the most lateral sections lack labels, one hemisphere lacks labels entirely, and label borders are jagged. Slight rotation of the underlying microscopy images leads to asymmetry between its two hemispheres. (**C**) Similar findings are apparent in the coronal view. (**D**) The Dice Similarity Coefficient (DSC), a measure of the completeness of labeling compared to the thresholded atlases, for the labeled hemispheres increased for all brains in the ADMBA after lateral edge extension (original median = 0.91, extended median = 0.93;

*Figure 2 continued on next page*

*Figure 2 continued*

p=0.02, Wilcoxon signed-rank test (WSRT); '*Mus musculus*' level −1, ID 15564 in the Allen ontology). (E) To fill in the lateral edges using existing labels, a representative lateral label plane was iteratively resized to fit the underlying microscopy images. The plane for each subsequent microscopy plane was thresholded, the bounding box extracted, and the labels resized to fit this bounding box, followed by conforming labels to the underlying gross anatomical boundaries (*Figure 2—figure supplement 2*). A stretch of compressed planes was expanded (*Figure 2—figure supplement 3*), and the completed hemisphere of labels mirrored to the other hemisphere after rotation for symmetry to complete the labeling. (F) Coronal view after the lateral edge extension.

The online version of this article includes the following figure supplement(s) for figure 2:

**Figure supplement 1.** Fraction of hemisphere that is unlabeled in each of the ADMBA atlases.

**Figure supplement 2.** Lateral edge extension pipeline.

**Figure supplement 3.** Expansion of compressed labels in a stretch of planes in the E18.5 atlas.

## Atlas similarity

To focus our analysis on the brain tissue, we excluded non-CNS tissue for the three embryos for which the brain was not dissected prior to imaging. Across all eight atlases the similarity between the microscopy images and the original label annotations was estimated by taking the threshold of each sagittal section on the left hemisphere of the brain as an approximation of ground truth and comparing the label coverage using the Dice Similarity Coefficient (DSC) (*Dice, 1945*), calculated by the Insight Segmentation and Registration Toolkit (ITK) (*Tustison and Gee, 2009*). Higher DSCs reflect greater similarity between the images and labels, with a maximum possible value of 1. The observed DSCs for the original labels in the annotated hemisphere ranged from 0.85 to 0.97 (median 0.91; *Figure 2*).

## Atlas extension

To generate labels across the entire brain, we extended the existing labels to the lateral edges by following histological boundaries, before mirroring the labels on the opposite hemisphere (*Figure 2*). The most lateral labeled sagittal section provided the initial seed from which to grow labels laterally. First, we resized this plane to match the extent of corresponding microscopy signal in the next, unlabeled lateral sagittal section (*Figure 2—figure supplement 2A,B*). Next, we refined label boundaries by eroding each label and regrowing it along histological boundaries, which we call 'edge-aware' refinement. To model these boundaries, we generated gross anatomical maps in 3D using edge-detection methods on the volumetric histology images. Taking the Laplacian of Gaussian (*Marr and Hildreth, 1980*) of each histological volumetric image highlighted broad regions of similar intensities, using a relatively large Gaussian sigma of 5 to capture only well-demarcated boundaries. A zero-crossing detector converted this regional map into a binary anatomical edge map (*Figure 2—figure supplement 2B*). After eroding each label, we next used this anatomical map to guide the regrowth of labels by a compact watershed algorithm (*Neubert and Protzel, 2014*) step, which adds a size constraint to the classic watershed algorithm. Distances of each pixel to its nearest edge map pixel formed anatomically based watershed catchment areas, while each eroded label served as a large seed from which filling of the nearest catchment areas began and grew until meeting neighboring labels. Thus, we extended a labeled sagittal plane to an unlabeled one, refined by the histology.

This process was repeated iteratively across all remaining lateral sections (*Figure 2—figure supplement 2B,C*). By using a 3D anatomical map for the watershed and seeding it with the prior plane's labels, we ensured continuity between planes. To model the tapering of labels laterally, we preferentially eroded central labels by weighting each label's filter size based on the label's median distance from the tissue section's outer boundaries. Central labels thus eroded away faster, while the next most central labels grew preferentially to fill these vacated spaces. In the E18.5 atlas, for example, this approach allowed the basal ganglia to taper off and give way to the dorsal and medial pallium (*Figure 2B*). Although edges between planes remained somewhat jagged, this extension step creates a substrate for further refinement.

After completing label coverage for the left hemisphere, both the labels and underlying microscopy images were reflected across the sagittal midline to cover the remaining hemisphere. Care was taken to ensure that the sagittal midline was identified correctly by inspecting and rotating the 3D

image and midline plane from multiple angles (*Figure 2C*). Recalculating the DSC between the microscopy images and labels for the left hemisphere showed greater similarity across all eight atlases with a median DSC improvement of 0.02 (p=0.02, WSRT) and a resulting DSC range of 0.87 to 0.97 (median 0.93, *Figure 2D*). Equivalent analysis of DSC for the whole brain would show substantial improvement due to the absence of labels on the right side in the original.

## Label smoothing

The ADMBA atlases have been provided as 3D volumetric images, combined computationally from the original 2D sagittal reference plates; 2D sections can be generated from these 3D images in orthogonal dimensions to the original sagittal view. Visual inspection of labels in the axial and coronal planes reveals high-frequency artifacts along most edge borders, likely from the difficulty of drawing contiguous borders in dimensions orthogonal to the drawing plane (*Figure 1*, *Figure 3*). To quantify the degree of label smoothness, we used the unit-less compactness metric (*Ballard and Brown, 1982*). The compactness measure applied in 3D incorporates both surface area and volume, allowing for quantification of smoothness to measure shape irregularity. Of note, compactness is independent of scale or orientation, facilitating comparison across all labels, despite size differences, and its sensitivity to noise allows finer detection of label irregularity (*Bribiesca, 2008*). Measuring the compactness of each label and taking the weighted mean based on label volume for each atlas gave a median compactness of 13,343 (mean: 34,099, standard deviation (SD): 44,988). For context, across all eight atlases the thresholded 3D whole-brain microscopy images were more compact (median compactness: 1895, mean: 2615, SD: 2305; p=0.02, WSRT, Bonferroni corrected for three comparisons; *Figure 3—figure supplement 1A*), consistent with the observed irregularity in the label images compared to anatomical images (*Figure 2B* and *Figure 3*).

To reduce this irregularity, we applied a smoothing filter iteratively to the 3D image of each label. Prior approaches to this problem used a Gaussian filter (kernel SD of 0.5, applied in two passes) (*Niedworok et al., 2016*). While this visually improved the smoothness of label edges, we observed sharp angles along the edges, presumably from the limitation of rounding blurred pixel values to the integers required for label identities rather than allowing the subtler floating-point gradients from the Gaussian kernel (*Figure 3—figure supplement 2A*). In addition, the Gaussian filter expanded the volume of each label, leading to label loss as larger labels enveloped smaller ones (*Figure 3— figure supplement 2B*).

Optimal smoothing would maximize the smoothness of each label (i.e. compactness; *Bribiesca, 2008*) whilst minimizing changes in shape and location. It would also not lead to label loss. To calculate the improvement in compactness, we defined 'compaction' as the difference in compactness between the original and smoothed label over the compactness of the original label with a range from 0 or 1, with 1 being optimal. For changes in shape and location, we defined 'displacement' as the fraction of the smoothed label that was outside of the original label with a range from 0 or 1, with 0 being optimal. We defined a 'smoothing quality' metric to reflect the balance of compaction and displacement, calculated as the difference between these two measures with a range from −1 or 1, with 1 being optimal. To estimate atlas-wide smoothing quality, we took the weighted sum by volume of smoothing quality for all labels in the atlas. Assessing the quality of smoothing using Gaussian blur with increasing Gaussian sigmas, we observed label loss in all atlases in the ADMBA, even with a small sigma where labels remained visibly jagged. At this sigma value, the median atlas-wide smoothing quality across all eight atlases was 0.54 (mean 0.53), rising to a peak of 0.57 (mean 0.56) at sigma 0.5 and 0.49 (mean 0.53) at a sigma of 1.0, but with substantial loss of labels (median 14% lost at sigma 0.25, rising to 42% lost at sigma 1; mean 11% and 47%, respectively *Figure 3—figure supplement 2B*).

To refine smoothing while minimizing label loss, we changed the filter from a Gaussian to a morphological opening filter (*Serra, 1983*). This filter first erodes each label to remove artifacts, followed by dilation to restore its original volume. To avoid label loss caused by excessive erosion of small labels, we halved the size of its structuring element for labels with ≤5000 pixels. A few labels were split into numerous tiny fragments that would disappear with an opening filter. For these small labels, the opening filter was replaced by a closing filter, reversing the process by dilating before eroding to reconnect these components. With this adaptive opening filter approach, labels became more compact with smoother edges, while retaining their overall shape as seen in both 2D and 3D visualizations (*Figure 3A*).

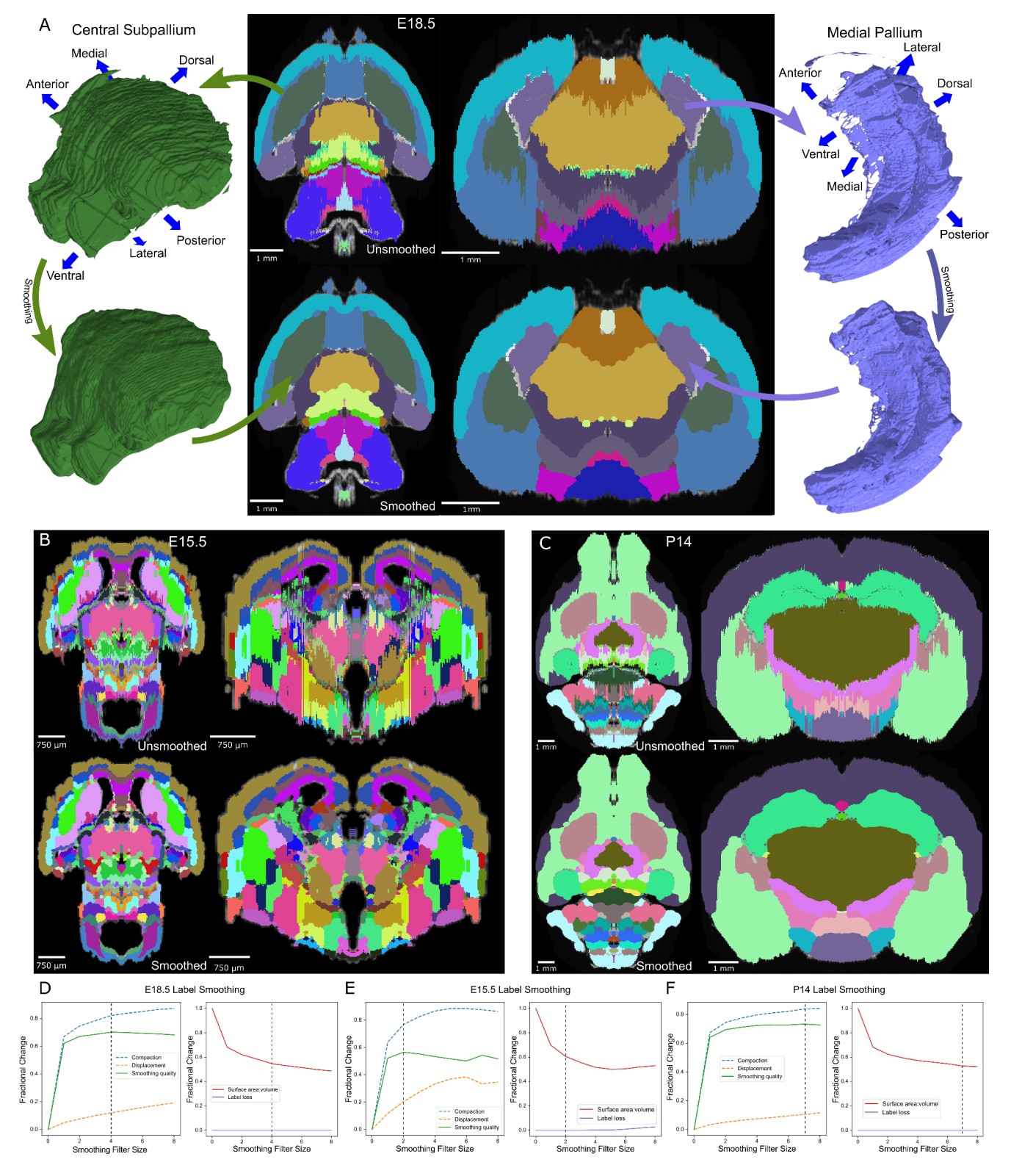

**Figure 3.** Atlas smoothing and optimization by adaptive morphological filtering. (**A**) Irregularities at the label edges in the original (mirrored) E18.5 atlas (top) were smoothed by applying an adaptive morphological opening filter iteratively to each label, starting from the largest to the smallest labels (bottom). 3D renderings (outer columns) are depicted for two representative labels before (top) and after (bottom) smoothing in the axial (middle left) and coronal (middle right) views. (**B, C**) The view in 'A' is repeated for E15.5 and P14, respectively. (**D**) To identify the optimal filter structuring element

*Figure 3 continued on next page*

*Figure 3 continued*

size, we devised an 'atlas-wide smoothing quality' metric to incorporate the balance between smoothing (compaction) and changes in size and shape (displacement). While compaction continued to improve with increasing structuring element size, displacement eventually caught up, giving an optimal atlas-wide smoothing quality with a structuring element size of 4 for the E18.5 atlas (left). We also assessed the number of labels that were lost (none in this example) and the surface area to volume ratio, for which lower values reflect smoother shapes (right). Vertical dashed lines indicate the optimal filter structuring element size, based on 'atlas-wide smoothing quality'. (E, F) This plot is repeated for E15.5 and P14, respectively.

The online version of this article includes the following figure supplement(s) for figure 3:

**Figure supplement 1.** Compactness of whole brain histology and labels.
**Figure supplement 2.** Smoothing labels by Gaussian blur.
**Figure supplement 3.** Smoothing quality across filter sizes for all ADMBA atlases.

## Smoothing quality across filter sizes for all ADMBA atlases

Quantifying the improvement with the adaptive opening filter approach, using only filter sizes that completely eliminated label loss, we obtained a median atlas-wide smoothing quality of 0.61 (mean 0.62) across all eight atlases, and improvement over the Gaussian filter approach (sigma 0.25; p=0.008; *Figure 3—figure supplement 2C*). The optimal filter size varied between atlases, ranging from 2 to 7 (E15.5 and P14 shown in *Figure 3C,D*; all ADMBA shown in *Figure 3—figure supplement 3*). The median overall compactness improved significantly (13,343 (SD = 44,988) for unsmoothed labels vs. 2527 (SD = 2634) for smoothed labels, p=0.02, WSRT, Bonferroni corrected for the three comparisons; mean 34,099 vs. 3172) to a level that did not differ from that observed for the microscopy images of whole brains (p=1.00, WSRT, Bonferroni corrected for the three comparisons; *Figure 3—figure supplement 1B,C*).

Because morphological filters such as erosion classically operate globally, a drawback to these filters is the potential loss of thin structures, such as loss of the thin portion of the alar plate of the evaginated telencephalic vesicle (*Figure 3A*). Smoothing in-place also does not address gross anatomical misalignment. We address these issues in subsequent steps.

## Label refinement by detected anatomical edges

The extending, mirroring, and smoothing steps lead to a more complete set of labels for the ADMBA and correct the irregular borders in orthogonal planes to which the original labels were drawn; however, in several locations the labels do not align closely to the anatomical edges seen in the underlying histology images, for example the basal ganglia do not follow the curve of the lateral septal nuclei (*Figure 2C*). To better map the anatomical fidelity of annotations in all dimensions, without manual relabeling, we leveraged our method for extending labels laterally based on gross anatomical edges to further refine all labels (*Figure 4*).

Using the same gross anatomical map in 3D (shown for an example plane in *Figure 4A*, second from left), we first quantified the distances from 3D label edges to the expected anatomical position. Assuming that the nearest gross anatomical edge was the correct one, we measured the distance from each label edge to the nearest gross anatomical edge. We can visualize this distance as a color gradient, in which higher intensity of color represents a greater distance to each anatomical edge (*Figure 4A*, right columns).

To modify the labels in light of the gross anatomical edge map, we again incorporated the edge-aware algorithm, this time in 3D. We made the assumption that the core of each label is annotated accurately, whereas label edges are more prone to inaccuracies from plane-to-plane misalignments or the difficulty of assessing histological edges. To preserve the core while reannotating the periphery, we first eroded each label to remove edges. These eroded labels became the seed for the watershed step, which re-grew labels back toward their original size but now guided by the gross anatomical edge map.

Normally, the erosion step would lead to loss of thin structures within labels because erosion operates globally on the entire label. To preserve these thin structures, we skeletonized each label in 3D, which thins the label to its core structure (*Lee et al., 1994*), and added the skeleton back to the eroded label. We used a much more lightly eroded version of the label for the skeletonization to avoid retaining label edges in the skeleton. By combining erosion with skeletonization when

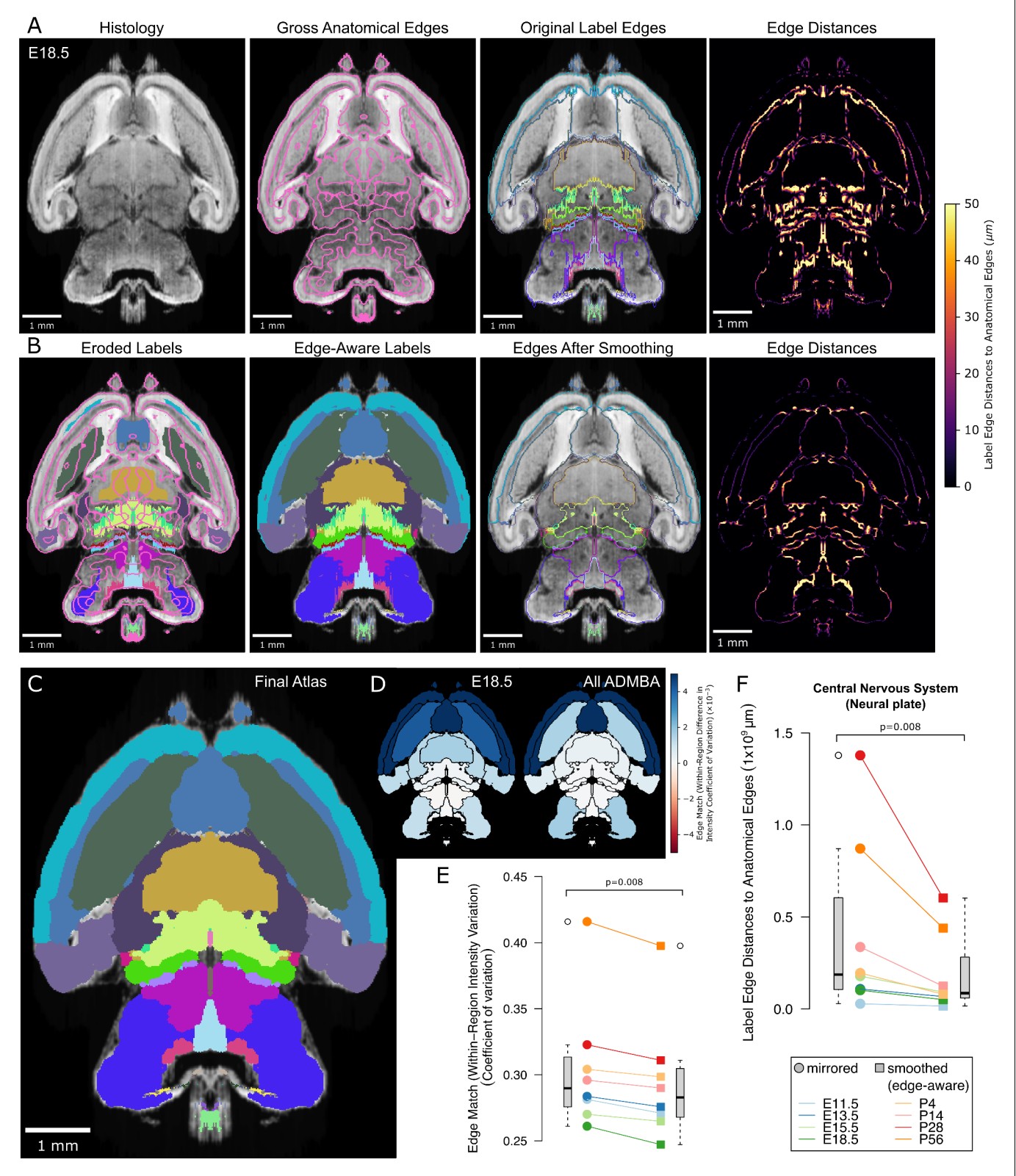

**Figure 4.** Edge-aware atlas reannotation based on underlying anatomical boundaries. (**A**) Edge detection of the volumetric histology image delineated gross anatomical edges (left), shown here with the E18.5 atlas. To compare these histology-derived anatomical edges with the extended and mirrored but unsmoothed label edges (center left), we used a distance transform method to find the distance from each label edge pixel to the nearest anatomical edge (center right), shown here as a heat map of edge distances (right). (**B**) Eroded labels served as seeds (left) from which to grow edge-

*Figure 4 continued on next page*

*Figure 4 continued*

aware labels through a watershed algorithm (center left), guided by the gross anatomical edges. After smoothing, borders matched the anatomical edges more closely (center right), as shown in the edge distance heat map for the modified labels (right), using the same intensity scale. (C) The final E18.5 atlas after edge-aware reannotation and label smoothing to minimize edge artifacts. (D) To evaluate the level of edge match by label, we mapped differences in the intensity coefficient of variation weighted by relative volume for each label before and after label refinement onto each corresponding label for the E18.5 atlas (left) and across all ADMBA atlases (right). For both, the anatomical map depicts this metric as a color gradient across all of the sublevel labels present in a cross-section of the E18.5 atlas. Improvements of this metric with the refined atlas are colored in blue, minimal change is shown in white, while red represents better performance with the original atlas. (E) Applied across the full ADMBA, edge-aware reannotation and smoothing led to a significant improvement in the overall variation of intensities, taken as a weighted mean across all labels to incorporate parcellation changes while weighting by label volume (central nervous system, or 'neural plate,' level 0, ID 15565 in the Allen ontology; p=0.008, n = 8 atlases, WSRT). (F) Distances from labels to anatomical edges taken as the sum across all labels similarly showed a significant improvement across atlases (p=0.008, n = 8, WSRT).

The online version of this article includes the following figure supplement(s) for figure 4:

**Figure supplement 1.** Edge distances with smoothing, edge-aware reannotation, or both.
**Figure supplement 2.** Volume changes by label from edge-aware smoothing across the ADMBA.
**Figure supplement 3.** Internal control using the asymmetric, contralateral hemispheres from the original atlases.
**Figure supplement 4.** Lateral extension comparison with partial annotations and ground truth.
**Figure supplement 5.** Edge-aware adjustment in delicate sub-regions with subtle contrast.
**Figure supplement 6.** Preservation of regions demarcated by in-situ hybridization markers.

generating seeds for the watershed, we retained thin structures such as the alar plate of the evaginated telencephalic vesicle located anterior to the basal ganglia.

After performing this edge-aware step, we ran the adaptive morphological opening filter smoothing step (*Figure 4B*). Because the edge-aware step partially smooths structures, we could use smaller filter sizes for smoothing to avoid loss of thin structures. The resulting labels show considerable improvement, for example the basal ganglia now curve around the lateral septal nuclei (*Figure 4C*). By adapting the morphological filter sizes during both the edge-aware and final smoothing steps, we avoid label loss and minimize volume changes relative to label size as seen in the smaller labels (*Figure 4—figure supplement 2*). Visualization of the color gradient of distances to anatomical edges also confirms substantial improvement in label alignment compared with the original labels or smoothing or edge-aware steps alone (*Figure 4—figure supplement 1B*). To quantify this improvement brain-wide, we calculated the sum of edge distances for each pixel at label surfaces across the ADMBA. We observed a significant reduction from a median of 187 million to 86 million µ*m* (p=0.008, WSRT; *Figure 4B,C,F*), with a median Dice Similarity Coefficient between original (mirrored) and smoothed (edge-aware) labels of 0.76 (mean 0.80) and 9% median (mean 12%) volume reassignment. Example planes from all atlases in the ADMBA before and after refinement are depicted in *Figure 5*, and movies across all planes are shown in *Figure 5—videos 1–16*.

Anatomical edges in microscopy images reflect differences in intensity between regions. Therefore, we would expect accurate labels to have smaller variation in intensity in the underlying microscopy images than inaccurate labels, although such a difference would need to be apparent even though the majority of the label is unchanged. We used the coefficient of variation of intensity values within each label to quantify this expectation and demonstrated significantly lower variation with edge-aware labeling (median from all labels weighted by size decreased from 0.290 to 0.282, p=0.008, for all eight atlases, WSRT; *Figure 4D,E*). Furthermore, edge-aware labeling decreased the absolute coefficient of variation for 92 of the 100 individual labels represented by all atlases. The few labels that showed increased variation were frequently in regions of relatively subtle intensity changes, such as the hindbrain, where histological edges were less well-defined (*Figure 4D*).

As an internal control, we also registered the atlases to their original, asymmetric microscopy images to compare the impact of atlas reconstruction in the originally labeled hemispheres and the contralateral, asymmetric hemispheres on which the atlases were not derived and found similar improvement in anatomical fidelity (*Figure 4—figure supplement 3*). We also compared the computationally generated labels and partially labeled lateral regions not included in the 3D reconstruction with manually annotated regions in the P28 atlas, which showed an increase in the Dice Similarity Coefficient in the reconstructed labels (*Figure 4—figure supplement 4*). To explore how the edge-aware approach handles delicate sub-regions, which may not have clearly demarcated anatomical

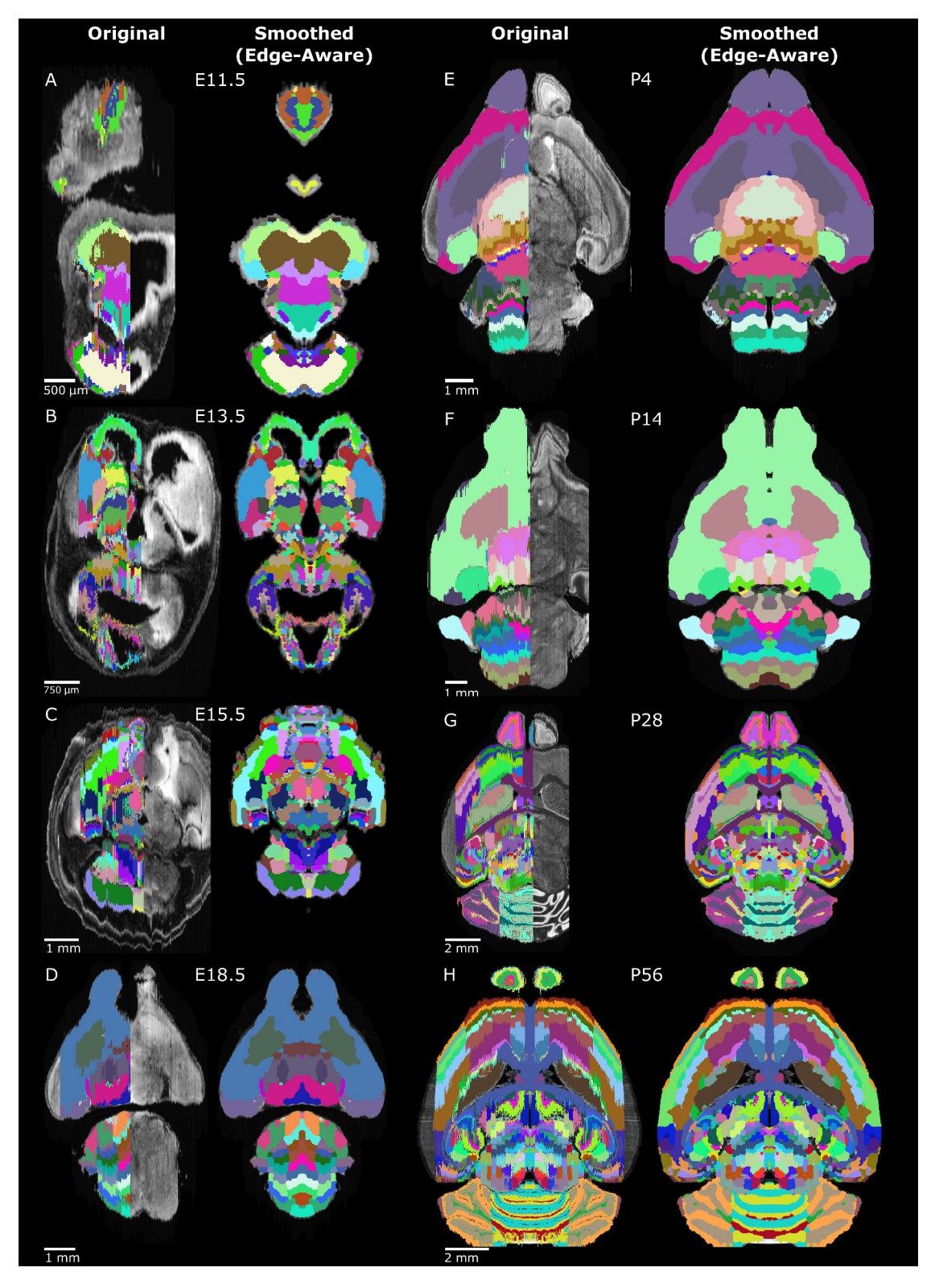

**Figure 5.** Original and 3D reconstructed atlas comparison across the ADMBA. (A-H) Representative axial planes from all atlases. For each pair of images, a plane of the original (left) atlas is depicted next to the refined (right) atlas after undergoing the full refinement pipeline. Complete atlases before and after refinement are shown as movies in *Figure 5—videos 1–16*.

*Figure 5 continued on next page*

*Figure 5 continued*

The online version of this article includes the following video and figure supplement(s) for figure 5:

**Figure supplement 1.** Piecewise 3D affine transformation of the spinal cord in the E11.5 atlas.

**Figure 5—video 1.** Original ADMBA E11.5 Atlas.
https://elifesciences.org/articles/61408#fig5video1

**Figure 5—video 2.** 3D reconstructed ADMBA E11.5 Atlas.
https://elifesciences.org/articles/61408#fig5video2

**Figure 5—video 3.** Original ADMBA E13.5 Atlas.
https://elifesciences.org/articles/61408#fig5video3

**Figure 5—video 4.** 3D reconstructed ADMBA E13.5 Atlas.
https://elifesciences.org/articles/61408#fig5video4

**Figure 5—video 5.** Original ADMBA E15.5 Atlas.
https://elifesciences.org/articles/61408#fig5video5

**Figure 5—video 6.** 3D reconstructed ADMBA E15.5 Atlas.
https://elifesciences.org/articles/61408#fig5video6

**Figure 5—video 7.** Original ADMBA E18.5 Atlas.
https://elifesciences.org/articles/61408#fig5video7

**Figure 5—video 8.** 3D reconstructed ADMBA E18.5 Atlas.
https://elifesciences.org/articles/61408#fig5video8

**Figure 5—video 9.** Original ADMBA P4 Atlas.
https://elifesciences.org/articles/61408#fig5video9

**Figure 5—video 10.** 3D reconstructed ADMBA P4 Atlas.
https://elifesciences.org/articles/61408#fig5video10

**Figure 5—video 11.** Original ADMBA P14 Atlas.
https://elifesciences.org/articles/61408#fig5video11

**Figure 5—video 12.** 3D reconstructed ADMBA P14 Atlas.
https://elifesciences.org/articles/61408#fig5video12

**Figure 5—video 13.** Original ADMBA P28 Atlas.
https://elifesciences.org/articles/61408#fig5video13

**Figure 5—video 14.** 3D reconstructed ADMBA P28 Atlas.
https://elifesciences.org/articles/61408#fig5video14

**Figure 5—video 15.** Original ADMBA P56 Atlas.
https://elifesciences.org/articles/61408#fig5video15

**Figure 5—video 16.** 3D reconstructed ADMBA P56 Atlas.
https://elifesciences.org/articles/61408#fig5video16

boundaries, we focused in greater depth on two complicated sub-regions in the P28 atlas (*Figure 4—figure supplement 5*) and two regions described in the original ADMBA paper (*Thompson et al., 2014b*) as demarcated by in-situ hybridization markers in the E13.5 and E15.5 atlases (*Figure 4—figure supplement 6*), showing that if anatomical data does not support moving the boundary, our approach smooths the boundary in the axial and coronal planes without moving the location in any of the planes.

## Application to tissue-cleared whole brains

A major goal of 3D atlas refinement is for automated label propagation to optically sectioned, volumetric microscopy images generated from intact tissue using recently developed tissue clearing techniques. As the accuracy of atlas registration is ultimately dependent on the fidelity of the underlying atlas in all dimensions, we sought to test and quantify improvement from our atlas refinements in cleared mouse whole brains.

Among the many available methods to clear whole organs, we chose CUBIC (*Susaki et al., 2014*; *Susaki et al., 2015*) given its balance of overall transparency, morphological retention with minimal tissue expansion, and ease of handling as a passive aqueous technique (*Kolesová et al., 2016*; *Xu et al., 2019a*; *Bossolani et al., 2019*). After clearing C57BL/6J WT mouse pup brains (age P0) with simultaneous incubation in SYTO 16 nuclear dye for 2 weeks, we imaged intact whole brains by lightsheet microscopy at 5x to obtain volumetric images at cellular resolution (*Figure 6—figure*

supplement 1), taking approximately 3 hr to image and generating approximately 500 GB of data per brain (n = 15; 10 male, 5 female).

To detect nuclei throughout cleared whole brains automatically, we implemented a 3D blob detection algorithm using the Laplacian of Gaussian filter, which has been shown to work well in equivalent image data (*Schmid et al., 2010*; *Figure 6*). To make the nuclei approximately spherical for blob detection, we interpolated the images axially to match the lateral resolution. Due to the large quantity of data, processing was performed in parallel on small chunks. Preprocessing and detection settings were optimized using hyperparameter tuning against a 'truth set' of 1118 nuclei selected from multiple brain regions that had been verified by manual visualization (*Figure 6—figure supplement 2A,B*). The resulting model achieved a recall (sensitivity) of 90% and precision (positive

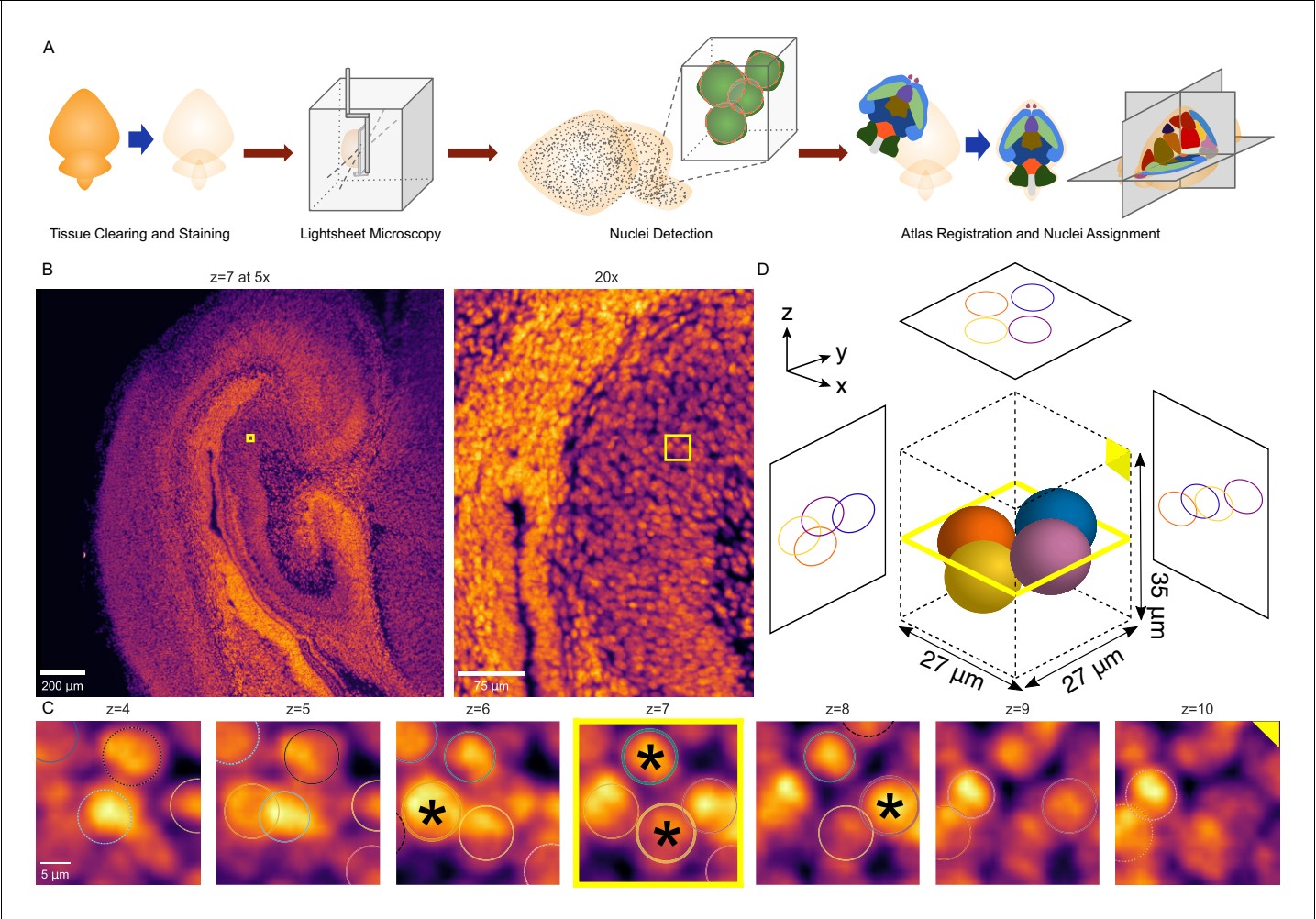

**Figure 6.** 3D whole-brain nuclei detection pipeline. (**A**) Overview of the nuclei detection and assignment pipeline. After tissue clearing by CUBIC and nuclear staining, we imaged intact whole P0 mouse brains by lightsheet microscopy (*Figure 6—figure supplement 1*). The E18.5 mouse brain atlas was registered to volumetric sample images. Nuclei were identified brain-wide with a 3D blob detector algorithm, and the nuclei coordinates were used to map to specific region labels, allowing quantification of nuclei counts and density for each label. (**B**) Axial view of the hippocampus of a P0 mouse brain at the original magnification of 5x (left) and zoomed to 20x (right), with the same region of interest (ROI) depicted by the yellow box in both. (**C**) Serial 2D planes along the ROI shown by the yellow box in part 'B' show the emergence and disappearance of nuclei while progressing vertically (z-axis) through the ROI. Each of four individual nuclei is assigned a unique colored circle. The most central image of the nuclei is indicated by a solid circle and asterisk, while images above or below have dashed circles. (**D**) 3D schematic of the four nuclei from part 'C' demonstrating their spatial orientation. The online version of this article includes the following figure supplement(s) for figure 6:

**Figure supplement 1.** Custom 3D printed specimen mount for suspending samples in the Zeiss Lightsheeet Z.1 microscope.

**Figure supplement 2.** Nuclei detection optimization and comparison with intensity.

**Figure supplement 3.** Registration of atlas to WT brains.

predictive value) of 92%. Furthermore, the model showed high correlation with total lightsheet microscopy intensity levels brain-wide (r > 0.99, p≤1 × $10^{-16}$, Spearman's rank correlation coefficient (SRCC) for both original (mirrored) and smoothed (edge-aware) atlases) and nuclei vs. intensity densities (original: r = 0.89; smoothed: r = 0.93; p≤1 × $10^{-16}$ for both, SRCC), suggesting the performance was accurate outside the narrow target of the truth set (*Figure 6—figure supplement 2C–E*).

Using the E18.5 brain as the closest atlas to our P0 C57BL/6J wild-type mouse brains (birth typically at E19 *Murray et al., 2010*), we then registered the atlas volumetric histology images to each sample brain. We chose to register brains using the Elastix toolkit (*Klein et al., 2010*; *Shamonin et al., 2014*), which has been validated on CUBIC cleared brains and balances computational efficiency and accuracy (*Nazib et al., 2018*). After downsampling the volumetric images to the same dimensions as the atlas for efficiency, we applied rigid, followed by non-rigid, deformable registration based on a multi-resolution scheme, optimizing the cross-correlation similarity metric as implemented by the SimpleElastix (*Nazib et al., 2018*; *Marstal et al., 2016*) programmable interface to the Elastix toolkit (*Klein et al., 2010*; *Hammelrath et al., 2016*). After registration, the median DSC between the registered atlas volumetric histology images and the lightsheet microscopy images of each sample brain was 0.91 (mean: 0.91; 95% confidence interval: 0.90–0.92; *Figure 6—figure supplement 3*), although inspection of registered brains also revealed slight misalignments with the atlas, mostly in caudal regions including the cerebellum, a structure known to be challenging for registration (*Nazib et al., 2018*). Variations in sample preparation, including the completeness of the dissected hindbrain and expansion of the third ventricle during tissue clearing may also have contributed to these misalignments.

To evaluate whether our updated, smoothed 3D labels improved the analysis of true volumetric data, we registered both the original and refined labels to these 15 tissue-cleared, nuclei-labeled whole wild-type mouse brains (10 male, five female). We would expect improved 3D labels to correspond more closely to the underlying anatomy. To test this expectation, we generated gross anatomical edge maps for each cleared brain and measured distances from label borders to anatomical edges. Edge distances significantly improved for almost all labels (overall decrease from a median of 153 million µm to 96 million µm, p=0.007, WSRT, Bonferroni corrected for all 120 labels across all hierarchical levels, mean 152 million to 99 million µm; *Figure 7A,C–D*; *Figure 7—figure supplement 1A*). We would also expect improved 3D labels to have lower variation in image intensity within each label, as we observed in assessing the refined labels with the original brain images from which they were derived (*Figure 4E*). We observed a small, but significant decrease in the intensity coefficient of variation at the whole-brain level (0.309 to 0.301, p=0.007, WSRT, Bonferroni corrected for the 120 comparisons, mean 0.311 to 0.304) and most sub-labels (*Figure 7—figure supplement 1B*). For 22 labels (18% of all labels) the variability worsened, however this was only significant (p=0.04) for a single label. The majority of these 22 labels describe small structures (combined 7% of total volume) and therefore sensitive to slight perturbations in border location, and located ventrally in the brain, where signal was prone to being distorted by glue artifacts from the mount for lightsheet imaging. For context, variability improved for 98 labels (82% of all labels) with 27 labels showing significant improvement.

These two analyses support an overall improved performance of the refined 3D labels with the volumetric images, however we can also test nuclei density - a key use case for whole-brain imaging methods (*Murakami et al., 2018*). After detecting nuclei throughout each brain (*Figure 6*), we assigned each nucleus to an atlas label based on the 3D location of the nucleus. Using the numbers of nuclei per label and the volume per registered label, we calculated nuclei densities using the original and refined labels (*Figure 7—figure supplement 2*). As with the volumetric image data, we would expect accurate labels to encapsulate regions with more constant nuclei density. We therefore assessed the coefficient of variation for nuclei density and observed a small, but significant improvement with the refined labels overall (median 0.629–0.625, p=0.007, WSRT, Bonferroni corrected for the 120 hierarchical labels, mean 0.629–0.625) and across the majority of labels in the hierarchy (*Figure 7—figure supplement 1C*). A median of 13.5% (4.7 million) nuclei (mean 13.2% [4.6 million] nuclei) were reassigned from original to refined labels. Examples of these nuclei reassignments in a wild-type mouse forebrain are depicted in *Figure 7B*. As an independent assessment of label alignment based on nuclei density alone, we measured nuclei clustering within each label under the expectation that well-aligned labels would group nuclei of similar densities, whereas poorly

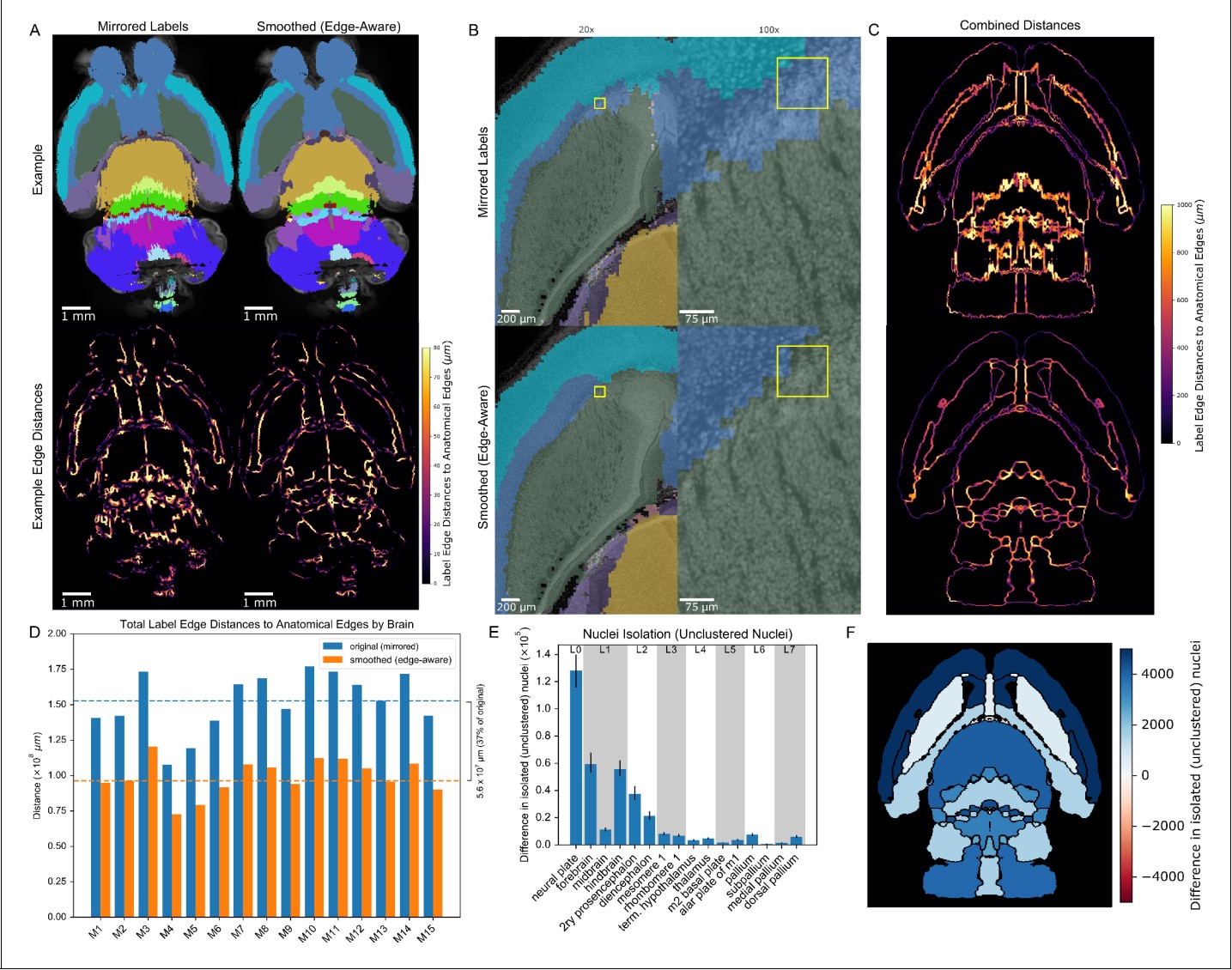

**Figure 7.** 3D reconstructed atlas validation in wild-type mouse brains at cellular resolution. (**A**) Example of the original (mirrored, left) and smoothed (edge-aware, right) E18.5 atlases registered to a representative cleared, optically-sectioned P0 wild-type mouse brain. Edge distances between the registered labels and the brain's own anatomical edge map are reduced for this example brain, shown by the color gradient for each edge (bottom). (**B**) Label alignment at higher resolution. The top row depicts the registered original (mirrored) atlas at 20x and 100x around a region of interest highlighted by a yellow box. This same brain region is depicted in the bottom row, but overlaid with the refined atlas registered using identical transformations as in the original atlas. (**C**) Summation of the edge distances across all 15 wild-type brains with color gradient showing the edge distances with the original (top) and smoothed (bottom) labels. (**D**) Total edge distance at the whole-brain level before and after atlas refinement for each of these brains. (**E**) Density-based nuclei clustering within each label. Many of the isolated nuclei that could not be clustered in the original labels were clustered in the refined labels, with differences in unclustered nuclei shown from 16 regions selected across the hierarchy of labels from the grossest (neural plate, L0, left) to the finest (e.g. dorsal pallidum, L7, right). Error bars represent 95% confidence intervals. (**F**) The differences between the original and refined labels' unclustered nuclei are depicted on an anatomical map showing this metric as a color gradient across all the sublevel labels present in a cross section. Improvements with the refined atlas are colored in blue, while red represents better performance with the original atlas. A complete list of differences for each metric in each label is provided in ***Supplementary file 2***.

The online version of this article includes the following figure supplement(s) for figure 7:

**Figure supplement 1.** Region homogeneity in wild-type brains based on the original vs.refined atlases.

**Figure supplement 2.** Changes in volumes, nuclei, and densities across wild-type brains based on atlas labels.

**Figure supplement 3.** Nuclei clustering in the original vs. refined atlases.

**Figure supplement 4.** MagellanMapper software GUI screenshots.

aligned labels would form isolated pockets of nuclei along borders between regions of contrasting nuclei densities. We employed Density-Based Spatial Clustering of Applications with Noise (DBSCAN) (*Ester et al., 1996*), an algorithm useful for both clustering and noise or anomaly detection (*Ali et al., 2010*), and indeed found that label refinement reduced isolated, unclustered nuclei by 30% (median $4.4 \times 10^5$ to $3.1 \times 10^5$ nuclei; p=0.007, WSRT, Bonferroni corrected for the 120 labels; mean $4.4 \times 10^5$ to $3.1 \times 10^5$; *Figure 7E–F*; *Figure 7—figure supplement 3*), suggesting greater nuclei assignment to labels with nuclei clusters of similar density.

## MagellanMapper as a tool

To facilitate both automated atlas refinement and visual inspection of large 3D microscopy images, we developed the open-source MagellanMapper software suite. This Python-based (*Oliphant, 2007*; *Millman and Aivazis, 2011*) image processing software consists of both a graphical user interface (GUI) for visualization and a command-line interface for automation (*Figure 7—figure supplement 4*). A major goal of the suite's GUI is to enable unfiltered, raw visualization and annotation of original 2D images in a 3D context. The GUI consists of three main components: (1) A region of interest (ROI) selector and 3D visualizer with 3D rendering, (2) A serial 2D ROI visualizer and annotator, tailored for building truth sets of 3D nuclei positions, and (3) A simultaneous orthogonal plane viewer with atlas label editing tools, including painting designed for use with a pen and edge interpolation to smoothly fill in changes between two edited, non-adjacent planes. The command-line interface provides automated pipelines for processing of large (terabyte-scale) volumetric images, including 3D nuclei detection at full image resolution.

The suite makes extensive use of established computer vision libraries such as scikit-image (*van der Walt et al., 2014*), ITK (*Yoo et al., 2002*) (via SimpleITK *Lowekamp et al., 2013* and SimpleElastix *Klein et al., 2010*; *Marstal et al., 2016*), visualization toolkits such as Matplotlib (*Hunter, 2007*) and Mayavi (*Ramachandran and Varoquaux, 2011*), and statistical and machine learning libraries such as Numpy (*van der Walt et al., 2011*), SciPy (*Virtanen et al., 2020*), Pandas (*McKinney, 2010*), and scikit-learn (*Pedregosa et al., 2018*). The cross-platform capabilities of Python allow the suite to be available in Windows, MacOS, and Linux. We also leverage additional open-source imaging ecosystems such as the ImageJ/Fiji (*Schindelin et al., 2012*; *Rueden et al., 2017*) suite for image stitching through the BigStitcher plugin (*Hörl et al., 2019*), integrated into our automated pipelines through Bash scripts.

## Discussion

3D reference atlases are critical resources for interpreting and comparing anatomically complex tissues, especially as techniques for whole-tissue microscopy (*Ragan et al., 2012*; *Murakami et al., 2018*) and in-situ single-cell transcriptome methodologies are refined (*Nowakowski et al., 2017*; *Polioudakis et al., 2019*). Considerable effort has been invested in generating 2D reference atlases, while manually building atlases in 3D is time-consuming and laborious; building 3D atlases from these 2D resources will save time and improve comparability between analyses. Here, we have presented a pipeline to automate the completion and conversion of atlases from 2D sections to complete, anatomically aligned 3D volumes and applied these methods to the eight developing mouse brain atlases in the ADMBA (*Figure 5*).

Key features of our pipeline include, first, an approach to adaptive morphology for label smoothing while preserving image detail by combining existing skeletonization, morphological erosion, and watershed segmentation methods and optimizing their parameters for application to 3D atlas reconstruction. Second, we utilized this method as an edge-aware label refinement approach, which incorporates anatomical edge detection from an underlying intensity-based image to perform two functions: (1) Label extension into incompletely labeled regions, and (2) Anatomically guided label curation. Third, we integrated these approaches into an automated pipeline as a general way to refine serial 2D labeled datasets into 3D atlases. We next applied the pipeline to complete and refine the full suite of 8 atlases in the Allen Developing Mouse Brain Atlas series from ages E11.5 to P56 as a resource to the neuroscience and neurodevelopmental communities (*Figure 5*). Specifically, these tools extended incomplete labels in each atlas to cover the entire brain and smooth artifacts between 2D sections, whilst respecting anatomical boundaries. We showed that the refined labels are more complete (*Figure 2*), the interface between labels match anatomical images more closely

(*Figure 4*), and the image intensity within these labels is more homogeneous (*Figure 4*). Using whole-brain lightsheet microscopy of 15 mice at P0, we identified the 3D location of 35 million nuclei per brain (median across 15 samples; *Figure 6*). We showed that the refined labels match anatomical boundaries better at cellular resolution, providing a key step towards comparative analysis of whole-brain cytoarchitecture (*Figure 7*). Fifth, we have provided the pipeline as an open source, graphical software suite with an end-to-end automated image processing pipeline for refining atlases, detecting nuclei, and quantifying nuclei counts by anatomical label in raw microscopy images of whole tissues.

Building a 3D reference atlas from a physically sectioned atlas presents three main challenges: (1) Incomplete labels resulting in unidentified tissues, (2) Smoothing jagged edge artifacts between sections in two of the three dimensions, and (3) Maintaining anatomical boundaries through the smoothing process. Prior work has tackled these challenges through two approaches. The first is automated smoothing of label boundaries (*Niedworok et al., 2016*); however, this approach has well recognized limitations (*Chen et al., 2019*), including labels crossing anatomical boundaries, loss of label detail, and complete loss of small labels (*Figure 3—figure supplement 2*). The second approach uses a combination of novel data generation and copious, manual curation as applied by neuroanatomists and illustrators to version 3 of the Allen Common Coordinate Framework (CCFv3, 2017) (*Allen Institute for Brain Science, 2017*; *Wang et al., 2020*). The resulting 3D reference atlas is highly accurate; however, an even greater large-scale effort would be required for the eight stages of development in the ADMBA.

Our method automates additional curation steps to extend and smooth labels guided by the underlying anatomy, providing an automated method closer to the accuracy of manual curation. To address loss of detail and loss of labels, we used the combination of adaptive morphology, an edge-aware watershed, and skeletonization to preserve small structures during automated smoothing. By fitting and extending labels to anatomical boundaries, we improved both the accuracy and label coverage of the resulting 3D reference atlas, without requiring additional data. While manual curation remains the gold standard, as performed in the Allen CCFv3, even this monumental effort translated only 77% of labels because of resource limitations (*Wang et al., 2020*). As suggested by Wang et al., future atlas development lies in automated, unbiased approaches (*Wang et al., 2020*). We demonstrate qualitatively (*Figure 5*) and quantitatively (*Figure 4*) that our automated approach is a substantial refinement over existing reference atlases and creates a complete resource that is immediately applicable for registration of 3D imaging data (*Figure 7*). Furthermore, it acts as a foundation for further improvements, substantially reducing the work required to generate a manually curated atlas in the future while freeing up anatomists and neuroscientists for the role of oversight in such endeavors.

Despite these substantial improvements, we are aware of some specific limitations of the approach. We expected less variability in image intensity within a region than between regions, and therefore improved region labels should reduce image intensity variability. While this expectation was correct for the majority of labels (92% of labels present in all atlases), a small number of regions showed a modest increase in variability with our refined labels. Most of these labels were in the hindbrain, where histological contrast is lower, leading to less well-defined anatomical edges. Adjusting the weight of edge-detected boundaries, based on the degree of underlying intensity contrast, could lessen the impact of these edges on reannotation. A few more labels exhibited this paradox when registered to the wild-type brains, likely owing to artifacts from tissue preparation and mounting, the slight age mismatch between the closest available E18.5 atlas and the P0 wild-type brains, registration misalignments, and inaccuracies of the blob detector.

Another limitation is the use of a single filtering kernel size for smoothing across the entire brain (*Figure 3*). In labels with both thick and thin portions, the thin portions are prone to being lost during the erosion step of smoothing. While a fully adaptive morphological approach would be beneficial, our use of skeletonization and the edge-aware watershed allowed us to preserve the core of such thin portions. More generally, the combination of erosion, skeletonization, and watershed could serve as an adaptive form of morphological operations to better preserve any label's overall footprint and topological consistency.

While the refined labels are superior to the original ones in both the ADMBA images and whole-brain lightsheet microscopy images, we observed more registration artifacts with the whole-brain lightsheet microscopy images (*Figure 7A*). Undoubtedly, biological variation, the slight age

mismatch (E18.5 in ADMBA vs. P0 for lightsheet), and the different preparations and imaging modality (Nissl sections vs. nuclei-stained whole brain) are substantial factors. Generating a group-wise composite of nuclei-stained whole brain could help reduce biological variation. Deferring the edge-aware reannotation step until after registration, using boundaries identified in the nuclei-stained whole brain, may also reduce these artifacts.

Recent methodological advances may further improve the refined 3D labels. *Chon et al., 2019* integrated the Franklin-Paxinos atlas with the CCFv3 atlas and additional cell-type specific markers, addressing discrepancies between these standard references. Integrating our anatomical edge maps with in-situ hybridization expression patterns (*Allen Institute for Brain Science, 2017*; *Thompson et al., 2014b*) may similarly improve label boundaries. A deep learning model utilized histological textures to generate labels (*Chen et al., 2019*). Training the model required substantial manual annotation to serve as the initial ground truth; in this context, our automated approach to atlases generation may be able to replace or augment this manual step. Another recent approach used fractional differential operators to preserve texture as well as edge information for segmentation of single-cell resolution data (*Xu et al., 2019b*). This could be incorporated into the 3D anatomical edge map generation step to further delineate labels at higher or even full resolution.

## Conclusion

Mouse whole-brain imaging has been used to understand models of human traits, including sexual dimorphism (*Kim et al., 2017*) and models of human disorders, including Alzheimer's disease (*Liebmann et al., 2016*; *Delafontaine-Martel et al., 2018*; *Whitesell et al., 2019*), serotonin dysregulation (*Ellegood et al., 2018*), epilepsy (*Intson et al., 2019*), and autism spectrum disorder (ASD) (*Suetterlin et al., 2018*). Such analyses would be augmented by accurate 3D reference atlases, allowing the detection of subtle quantitative changes not readily appreciable in individual slices. As we have described and demonstrated here, anatomically guided completion and refinement of reference atlases more fully leverages the many existing and established atlases by expanding them to full coverage and greater accuracy at cellular resolution. The completed and refined 3D ADMBA serves as a resource to help identify biological differences in the multiple models of human disorders and traits across brain development.

# Materials and methods

**Key resources table**

| Reagent type (species) or resource | Designation | Source or reference | Identifiers | Additional information |
|---|---|---|---|---|
| Genetic reagent (*Mus. musculus*) | C57BL/6J | Jackson Laboratory | Stock #: 000664 RRID:MGI_3028467 | |
| Commercial assay or kit | SYTO 16 | Thermo Fisher Scientific | Cat. #: S7578 | 1 $\mu M$ |
| Software, algorithm | FIJI/ImageJ software | FIJI/ImageJ | RRID:SCR002285 | Version 1.52 |
| Software, algorithm | MagellanMapper software | This paper | | |
| Software, algorithm | Zeiss Zen software | Zeiss | RRID:SCR_018163 | Version 2014 |

## Original atlases

The ADMBA series provides atlases for multiple developmental time points from embryonic through adult stages (*Thompson et al., 2014b*). Each atlas consists of two main images given in a volumetric format (.mhd and its associated .raw file), a microscopy image ('atlasVolume') and a labels image ('annotation'), which are each composed of multiple 2D sagittal planes. As outlined in the ADMBA technical white paper (*Allen Institute for Brain Science, 2013a*) and online application programming interface (API) documentation (*Allen Institute for Brain Science, 2018*), each microscopy plane is from imaging of a C57BL/6J mouse brain specimen cryosectioned into 20 $\mu m$ (E11.5-P4) or 25 $\mu m$ (P14-P56) thick sagittal sections stained with Feulgen-HP yellow nuclear stain (E11.5-E18.5) or Nissl staining (P4-P56), and imaging planes were assembled into a 3D volume. To create the atlas labeling, an expert anatomist used Adobe Illustrator CS to annotate 2D planes, which were

interpolated to create annotations in 3D (*Thompson et al., 2014b*; *Allen Institute for Brain Science, 2013b*; *Allen Institute for Brain Science, 2018*). The annotation planes correspond to each microscopy plane, with integer values at each pixel denoting the label at the finest structural level annotated for the corresponding microscopy voxel.

## Lateral labels extension

While the ADMBA covers a large proportion of unique areas within each brain, the atlas labels leave the lateral edges of one hemisphere and the entire other hemisphere unlabeled. To fill in missing areas without requiring further manual annotation, we made use of the existing labels to sequentially extend them into the lateral unlabeled histology sections. This process involves (1) resizing the last labeled plane to the next, unlabeled plane, (2) refitting the labels to the underlying histology, and (3) recursively labeling the next plane until all unlabeled planes are complete (*Figure 2—figure supplement 2*).

To extend lateral edges of the labeled hemisphere, we first identified the lateral-most labeled plane of the atlas microscopy images. We started from the first sagittal plane on the labeled side of the image and moved toward and into the brain, checking each plane for the presence of any label. Once we identified a contiguous stretch of planes with labels, we used the most lateral labeled plane as the template for subsequent labels to be extended out laterally, in the opposite direction. In a few atlases (e.g. P28), the lateral-most labeled planes are only partially complete, in which case the most lateral completely labeled plane was manually specified instead.

This last labeled lateral plane contained one or more discrete structures to extend label coverage, typically the cortex and sometimes the cerebellum. To find each structure and its associated labels, we first generated a mask of the histology by taking a slightly dilated version of the labels (morphology.binary_dilation method in scikit-image, typically with a disk shaped structuring element of size 5) to capture nearby unlabeled histological structures, thresholded this histology plane by a value of 10, removed small objects of size less than 200 connected pixels (default connectivity of 1 in morphology.remove_small_objects), and identified bounding boxes of connected histology components (measure.regionprops) and their matching labels. These bounding boxes contain the discrete structures that we will follow laterally, using the corresponding labels for each structure as templates to extend into the rest of each structure.

After identifying each structure and its labels, we fit the labels to the next lateral plane and recursively generated a new template for the subsequent plane. To fit the labels to the next plane, we found the histology bounding box for each structure and resized its labels to this box. We assumed for simplification that each structure is the same or smaller size in each subsequent plane, such as the tapering profile of the cortex laterally, and took only the single largest object found within the structure's bounding box.

Spline interpolation, anti-aliasing, and range rescaling were turned off during the resize operation to avoid introducing new label values. Some atlas labels contain empty space such as ventricles. To ensure that the ventricles close as they progress laterally, we employed an in-painting approach. Using the Euclidean distance transform method from the Scipy (*Virtanen et al., 2020*) library (ndimage.distance_transform_edt), we identified the indices of the nearest neighboring pixel for any unlabeled pixel whose corresponding histology intensity value was above threshold and filled this missing pixel label with the value of this neighbor.

## Edge map generation

While the labels from one plane could serve as an approximate template for the next plane, we curated this template to fit the underlying anatomy. To map this anatomy, we generated gross anatomical edge maps of the histology images through 3D edge detection. First, we smoothed the volumetric microscopy image using a Gaussian filter with a sigma of 5 followed by an edge detection with a Laplacian filter using a default operator size of 3, using both filters implemented in scikit-image (filters.gaussian and filters.laplace, respectively). This relatively large Gaussian sigma allowed for capture of broad anatomical edges such as the cortex and basal ganglia while minimizing detection of artifactual boundaries. To enhance edge detection of the outermost boundaries, we identified and removed background by thresholding the original microscopy image with an Otsu

threshold (filters.threshold_otsu) and combining it with a mask of all the original labels to fill in any missing holes in the thresholded image.

Finally, we reduced the edge-detected image to a binary image of edges alone by applying a zero-crossing detector. This detector separately erodes and dilates (morphology.erosion and morphology.dilation, respectively, with a ball-shaped structuring element of size 1) the edge-detected image to find borders, taking all pixels where this image changed signs as gross anatomical edges (*Figure 4A*).

## Anatomically guided serial 2D reannotation

This edge map allowed us to conform labels to local anatomy. To refit labels, we eroded and regrew them in a watershed transformation guided by the anatomical edge map.

First we eroded labels individually (morphology.binary_erosion in scikit-image with a ball-shaped structuring element of manually determined radii for each atlas). These eroded labels served as seeds for a compact watershed (morphology.watershed implemented in scikit-image with a compactness parameter of 0.005) (*Neubert and Protzel, 2014*; *Kornilov and Safonov, 2018*), guided by the anatomical edge map. We used the Euclidean distance transform of the anatomical edge map to define the catchment basins for the watershed transformation, where voxels farther from anatomical edges are at the bottoms of basins and fill faster, guiding the regrowth of eroded labels. Labels typically crossed several anatomical edges but tended to meet neighboring labels at common edges (*Figure 4B*). To limit the watershed to atlas foreground, which prevents label spillover across empty spaces, we set the watershed mask parameter to the original total labels foreground smoothed by an opening filter (morphology.binary_opening with a ball structuring element of radius 0 [off] to two determined for each atlas to avoid label loss and minimize volume changes) to remove artifacts around small ventricular spaces that might otherwise be crossed. The eroded labels thus regrew to fit anatomical guides and became the new, anatomically refined template to extend labels into the next plane.

To model the tapering and disappearance of labels laterally, we allowed labels to erode completely. We weighted erosion toward central labels within each structure by multiplying the erosion filter size for each label by the label's median distance to the structure perimeter divided by the maximum distance. Instead of simply eroding away the smallest labels, this approach preferentially eroded labels farthest from the perimeter, typically central labels.

In atlases such as the ADMBA E18.5 atlas, a few spurious labeled pixels from other regions regrew to create artifacts. To filter these spurious pixels, we applied smoothing to the initial labels template using the smoothing approach described below. Also, we applied skeletonization as outlined below to avoid loss of thin structures during erosion.

Each plane of labels thus conformed to its underlying anatomy and became the template for the next plane, keeping labels inherently connected from one plane to the next. While this approach is in serial 2D rather than fully 3D because the starting labeled plane is 2D, a subsequent step will further refine labels in 3D.

## Atlas 3D rotation and mirroring

To fill the missing hemisphere in the labels image, we initially simply mirrored the present labels and underlying histology planes across the first unlabeled sagittal plane on the opposite side. We noticed, however, that this mirroring frequently duplicated midline structures because the labels extend slightly past the true sagittal midline. When we shifted the mirroring to the sagittal plane closest to midline, we found that many labels were lost in the final image.

Many of the developing atlases contain at least a few labels positioned solely across the midline in the otherwise unlabeled hemisphere. While it is possible that these labels represent structures unique to one hemisphere, at least some of these labels are on the other side of the midline in other atlases of the ADMBA and thus more likely represent artifact. In some cases (e.g. P4 and P14), mirroring just slightly past the sagittal midline preserved these labels, whereas other atlases (e.g. P28) contained over 100 labels past the midline. One approach to preserve these labels would be to compress the near-midline labels to bring all labels into a single hemisphere at the expense of potentially misaligning otherwise appropriately positioned labels. To avoid this side effect, we elected to cut off a few labels to preserve the placement of the majority of labels.

Upon closer inspection, many of the brains are slightly rotated in two or three dimensions, which contributed to loss of midline labels during mirroring. To rotate images volumetrically in 3D, we applied the scikit-image rotate function to all 2D planes along any given axis for each rotation. For each atlas, we applied this volumetric rotation along all necessary axes until the sagittal midline was parallel to an image edge, manually inspecting each brain in our 2D/3D orthogonal viewer to ensure symmetry. In some cases, such as the E18.5 atlas, the brain skewed laterally along its rostrocaudal axis. To avoid introducing a gap between midline labels and the corrected midline after rotation, we filled all planes on the unlabeled hemisphere side with the last labeled plane before rotation. Mirroring would then overwrite all of these repeated labels except those that filled in potential gaps left by rotation. After rotation, we noticed that mirroring reduced label loss in at least some atlases (e.g. E13.5).

## Piecewise 3D affine transformation

The distal spinal cord of the E11.5 atlas is strongly skewed and would be duplicated during mirroring. The skew is complicated by the cord's coiling back on itself and progressive skew along its length. To bend the distal cord to the sagittal midline, we developed a method for piecewise 3D affine transformation. This transformation allows a cuboid ROI within a volumetric image to be sheared while maintaining a specified attachment point along another axis, reducing discontinuity with neighboring areas.

First, we specify an axis along which to shear and the degree of shearing for a given ROI. Each full plane along this axis is shifted in the indicated direction by a progressively larger amount, overwriting pixels into which the plane is shifted and filling vacated pixels with background to shear the stack smoothly in 3D. If an axis of attachment is also specified, each plane is sheared line-by-line along this axis so that the resulting parallelogram remains fully connected at one end of each axis. The resulting ROI is thus sheared in 3D while remaining smoothly connected to its surrounding space along two orthogonal faces of the original cuboid ROI to minimize disruption.

Applied to the skewed spinal cord, this approach allowed us to shear the cord one section at a time as it curved back along itself (*Figure 5—figure supplement 1*, left column), with the connected faces ensuring that each piece remains attached to one another. First we sheared the entire distal cord from the start of the skew (*Figure 5—figure supplement 1*, middle left column). This shift brought the proximal section toward midline, but the more distal cord remained skewed where it curved back on itself. To correct this skew, we sheared again but starting from a more distal point and along an orthogonal angle to follow the cord's curve (*Figure 5—figure supplement 1*, middle right column). Now most of the cord lined up with the sagittal midline, but the shear exacerbated the skew of the most distal cord. Finally, we applied a third affine, this time on only the most distal section and in the opposite direction as the prior affine (*Figure 5—figure supplement 1*, right column).

After mirroring the resulting brain and cord, no duplication could be seen. This piecewise, overlapping affine of targeted regions allowed straightening of the cord without breakages or alteration of surrounding areas. Although surrounding non-CNS tissue suffered noticeable breakage, they were stripped out as described below.

## Stripping non-CNS signal

The extended labels allowed us to mask and crop out non-CNS tissue, including the rest of the embryo present in several of the embryonic stage atlases. While distinguishing this tissue based on intensity characteristics alone would be challenging, especially for areas where the spinal cord extends the length of the embryo, the extended, mirrored labels provide a map of relevant CNS tissue.

For each of the atlases with non-CNS tissue (E11.5-E15.5), we first cropped the atlas to the bounding box of these labels along with a small padding (5px) to remove much of the non-CNS tissue, such as the entire unlabeled body in the E15.5 atlas. We removed non-CNS tissue remaining within the cropped areas by using the labels as a mask to remove all histology pixels outside of the labels mask, including the body surrounding the labeled spinal cord in E11.5. To avoid missing tissue that may have been unlabeled, we dilated the labels mask slightly (morphology.binary_dilation with a ball-shaped structuring element of size 2) so that the mask encompasses both the labels and its

immediate surroundings. The resulting histology thus contains all pixels in the near vicinity of labels, including pixels that should be labeled but are not.

## Label smoothing

While labels appear generally smooth when viewed from the sagittal plane, label edges are noticeably jagged when seen from the orthogonal directions. A previous smoothing solution proposed by *Niedworok et al., 2016* utilized a Gaussian blur with a sigma of 0.5 in two iterations to minimize ragged edges in an atlas derived from the Allen Reference Atlas P56 mouse brain atlas. We applied this approach by iteratively applying the Gaussian filter implemented in scikit-image (filters.gaussian) with a range of sigmas (0.25–1.25) to each label in 3D (*Figure 3—figure supplement 2*).

To extract each label in 3D, we found the bounding box of the label using the scikit-image measure.regionprops method and added additional empty padding space around the box. The Gaussian filter was applied to the label, and the original label in the bounding box was replaced with the smoothed label. Each label smoothing left small gaps vacated by the previously ragged borders. To fill in these gaps, we employed the in-painting approach described above. Finally, we replaced the original bounding box with that of the smoothed label in the volumetric labels image. We repeated the process for all labels from largest to smallest to complete the smoothing.

To enhance smoothing while retaining the original underlying contour of each label, we devised an adaptive opening morphological filter approach. The morphological opening filter first erodes the label to remove artifacts such as ragged edges, followed by dilation of the smoothed label to bring it back toward its original size. In place of the Gaussian filter, we applied this opening filter (morphology.binary_opening in scikit-image). For smaller labels, the filter occasionally caused the label the disappear, particularly for sparsely populated labels with disconnected pixels. To avoid label loss, we employed an adaptive filter approach by halving the size of the filter's structuring element for small labels ($\leq$5000 pixels). For any label lost in spite of this filter size reduction, we switched the filter to a closing morphological filter (morphology.binary_closing), which dilates first before eroding and tends to preserve these sparse labels at the expense of potentially amplifying artifact.

## Smoothing quality metric

To evaluate the quality of smoothing and to optimize morphological filter structuring element sizes, we developed a smoothing quality metric with the goal of balancing smoothness while maintaining the overall original shape and placement. Since the major effect of label smoothing is to minimize high frequency aberrations at label borders and thus make each label more compact, we measured this amount of smoothing by the 3D compactness metric. We termed the difference in compactness before and after smoothing as 'compaction.' As the goal of smoothing is to remove these artifacts while preserving the overall shape of the label, we introduced a penalty term of 'displacement,' measured by the volume shifted outside of the label's original bounds.

We used the classical measure of volumetric compactness, where lower values are more compact (*Ballard and Brown, 1982*):

$$\mathrm{Compactness} = \mathrm{SA}^3/\mathrm{Vol}^2 \qquad (1)$$

To measure the surface area of each label in 3D, we employed the marching cubes algorithm (*Lorensen and Cline, 1987*) as implemented in scikit-image (measure.marching_cubes_lewiner), which also accounts for anisotropy. The algorithm extracts surfaces in 3D by dividing the volume into cubes and marching through these cubes to find where isosurfaces intersect with each cube, forming triangular surfaces within each cube wherever an isosurface passes through the cube. Using a mask of each label, we obtained the edge surface as a mesh from which we measured the surface area (measure.mesh_surface_area). We took the volume as the total number of mask pixels and multiplied by the product of the voxel spacing to account for anisotropy. With the surface area and volume, we calculated the labels compactness (1). To quantify the fractional change in compactness before and after smoothing, we took the original compactness minus the smoothed compactness and normalized the difference to the original compactness to give the unitless value that we termed 'compaction':

$$Compaction = \frac{C_{orig} - C_{smooth}}{C_{orig}} \tag{2}$$

where C is the 3D compactness given above.

To measure 'displacement,' we measured the volume shifted outside of the label's original bounds. Taking the mask of the smoothed label, we combined it with the inverse of the mask of the original label through a boolean 'and' before totaling the number of pixels. Similarly to the compactness measure, we normalized the displacement volume to generate a unitless value constrained between 0 and 1, dividing this displaced smoothed volume by the total smoothed volume:

$$Displacement = \frac{Vol_{smooth} \notin Vol_{orig}}{Vol_{smooth}} \tag{3}$$

where Vol is the volume of the given label.

As a measure of smoothness quality, we took the difference of the compaction and displacement:

$$Smoothing\,quality = Compaction - Displacement \tag{4}$$

To quantify the smoothing quality for the entire atlas, we took the weighted arithmetic mean of all the labels' smoothing qualities, weighting by each label's volume. The atlas-wide smoothing quality metric can be summarized in the following equation:

$$Atlas - Wide\,Smoothing\,Quality = \frac{\sum\limits_{i=1}^{N} SmoothingQuality_i\,Vol_{i,orig}}{\sum\limits_{i=1}^{N} Vol_{i,orig}} \tag{5}$$

where N is the total number of original labels. While maximizing compaction would reduce surface area the most by transforming the label into a perfect sphere, the displacement from the label's original space would penalize this over-compaction, allowing us to target the balance of compaction and displacement to find the optimal smoothing quality.

## Anatomical to label edge distance quantification

As an automated method of quantifying the correspondence between labels and anatomical edges, we used the anatomical edge maps generated earlier to measure the distance between anatomical and label edges. We reduced each label to its edges in 3D by eroding the very outer surface of each label (morphology.binary_erosion with a cross-shaped structuring element with connectivity of one) and subtracting this eroded label from the original label.

To measure distances between label and anatomical borders, we performed the Euclidean distance transform (ndimage.distance_transform_edt in Scipy) on the anatomical edge map to measure distances from any given voxel to the anatomical edges, using the sampling parameter to specify the microscopy image spacing in μm. Using the labels edge map, we next took only the voxels in the distance map corresponding to these label borders as a map of distances from each label voxel to its nearest anatomical edge. As overall measures of distance from label borders to the nearest anatomical border, we summed the edge distances for each label to compare before and after label refinement.

## Anatomically guided 3D reannotation

Generating edge images provided a map of the boundaries between anatomically distinct regions not only to measure distances between borders, but also to curate the labels themselves with these anatomical edges. To reannotate the existing unsmoothed labels, we eroded and regrew them through an anatomically guided watershed transformation similar to the approach described above but now in 3D. We tested seeds with multiple erosion filter sizes and found that a structuring element size of 8 reduced the seed size sufficiently to correct label bounds around several grossly abnormal labels, including the lateral septal nuclei and basal ganglia.

As a global operator, erosion typically leads to loss of thin structures, especially with larger structuring elements. To avoid this loss, we first extracted the core structure of each label by finding its

3D skeleton (morphology.skeletonize_3d in scikit-image). We added the skeletonized image back to the eroded label to recover the location of thin structures, allowing the watershed to regrow these labeled areas in addition to the eroded label. To limit branches in the skeleton, which could counter the effect of the erosion, we input a lightly eroded version of the labels (structuring element half the size of that for the main erosion) to the skeletonization.

The resulting watershed segmentation also served as preliminary smoothing but introduced its own label edge artifacts, though of lower frequency than in the original labels. We thus deferred the smoothing algorithm until after this watershed step and could use smaller filter sizes to generate a final smooth image.

### Application to the full ADMBA series

Each atlas in the ADMBA required a different set of refinement features and settings, including specialized adjustments such as the 3D piecewise affine only for a specific atlas (E11.5). To allow for customized settings, we defined separate profiles of parameters within our software suite for each atlas.

### E11.5

The microscopy images depict a specimen in embryonic stage with the caudal end including the spinal cord wrapping around itself and deviated laterally from the rest of the body. As with most other atlases in this series, one half of the labels were missing. Making the image symmetric would allow us to mirror labels from the existing side onto this missing side and minimize bias when using the atlas for automated registration tasks. We started by rotating the image by 5° in the axial planes toward the left of the brain and 1° in the coronal planes to raise the right side of the brain, bringing the sagittal midline parallel to an image edge. To shift the distal end of the spinal cord back toward the sagittal midline, we applied the piecewise 3D affine transformation described above (*Figure 5—figure supplement 1*). Prior to the affine, we applied an additional rotation of 30° in the sagittal planes to position the skewed distal cord within a single cuboid ROI parallel to the image.

With the embryo now symmetric, we mirrored the labels and microscopy signal along the embyro's midline, measured as 52% across the sample along the z-axis. To highlight the central nervous system and remove breakages introduced by the affine transformations in non-CNS tissue, we stripped out non-CNS tissue as described earlier.

The E11.5 atlas uniquely contains labeled ventricles. To ensure that the Laplacian of Gaussian edge-detection algorithm appropriately found ventricular edges, we used only the thresholded atlas rather than incorporating the labels to find the background. We also included the ventricular space as foreground for purposes of the DSC calculations between microscopy and labels to account for the ventricular labeling. This atlas did not require lateral extension.

### E13.5

This atlas also contains the full embryo, but the spinal cord is symmetric along the sagittal midline of the brain and did not require the affine transformations as in the E11.5 atlas. After extending the lateral edges, we rotated the atlas by 4° in the axial planes toward the left of the brain and 2° in the coronal planes to lift the right side of the brain, making the atlas symmetric before mirroring the atlas at 48% along the sagittal planes. We again stripped non-CNS tissue the same way as for the E11.5 atlas.

### E15.5

This atlas is the final one to include the complete embryo, although only the very rostral end of the spinal cord includes labels. We extended the lateral edges, rotated the images by 4° in the axial planes toward the left side of the brain, mirrored the atlas at 49% along the sagittal planes, and stripped away the entire embryo outside of the brain. A stepwise shift in sagittal planes is apparent at several planes (e.g. 104 and 129) in both the histology and label images in the original atlas, which we smoothed slightly in the labels during the smoothing step.

## E18.5

A small subset of labels from sagittal planes 103–107 were compressed along the dorsoventral axis. To match them with their neighboring label planes and the underlying atlas, we resized them similarly to the lateral edge extension. Starting with the first plane in this subset, we thresholded the microscopy image, removed small objects, and obtained the largest bounding box of connected histology components. Taking only this largest connected structure allowed us to avoid including extraneous tissue visible on the ventral aspect of the brain, which was unlabeled. We repeated the process on the labels to obtain its compressed bounding box and resized it to the size of the microscopy bounding box. Finally, we repeated the entire process on the rest of the planes in this subset (*Figure 2—figure supplement 3*).

For the lateral edge extension, we noted that the basal ganglia in the most lateral planes are slightly larger than in more medial planes. Under the assumption that the basal ganglia would be tapering laterally, we skipped these planes and started the extension at 13.7% along the sagittal planes to take the plane with smallest basal ganglia label. After rotating the atlas by 1.5˚ in the axial planes toward the right of the brain and 2˚ in the coronal planes to lift the left side of the brain, we mirrored microscopy and labels at 52.5% along the sagittal planes for symmetry.

## P4

This atlas is reminiscent of E18.5 but without the necessity of setting the lateral edge extension starting plane explicitly or expanding any compressed labels. After lateral edge extension and rotating the atlas by 0.22˚ in the axial planes toward the right of the brain, we mirrored microscopy and labels at 48.7% along the sagittal planes for symmetry.

## P14

The most laterally labeled plane is discontinuous with the rest of the labeled planes in this atlas, so we skipped this plane during lateral extension and started extending only from the first contiguous set of planes. After rotating the atlas by 0.4˚ in the axial planes toward the left of the brain, we mirrored the microscopy and labels images at the 50% mark along the sagittal planes.

## P28

The most lateral planes had incomplete labels, requiring use of a more medial plane with complete labels at 11% along the sagittal planes for the lateral edge extension. After rotating the atlas by 1˚ in the axial planes toward the right of the brain, we mirrored the microscopy and labels images at the 48% mark along the sagittal planes.

## P56

The ADMBA contains a P56 mouse similar to the adult P56 but following the same ontological labeling scheme as in the rest of the ADMBA. This atlas uniquely contains bilateral labels, although the far lateral section is still missing. We again extended the lateral edges and mirrored the microscopy and labels images, starting extension at 13.8% and mirroring at 50% along the sagittal planes, respectively. The most lateral labeled plane contains two distinct labeled structures, the cortex and cerebellum, requiring separate extension for each distinct structure as outlined in our method above.

## Animals and tissue clearing

All procedures and animal care were approved and performed in accordance with institutional guidelines from the University of California, San Francisco Laboratory Animal Research Center (LARC). All strains were maintained on a C57BL/6J background. Animals were housed as one breeding pair per cage in a vivarium with a 12 hr light, 12 hr dark cycle. For timed pregnancies, noon on the day of the vaginal plug was counted as embryonic day 0.5. Pups were harvested at P0 (postnatal day 0). For assessing the eight atlases in the ADMBA, sample size was determined by the number of atlases available in the original 2D resource (n = 8). The size of our validation cohort (P0 wild-type mouse brains, n = 15) was chosen to detect an effect size of 1.5 after correction for multiple comparisons using a power calculation based on a paired t-test.

At the time of experiment, P0 pups were anesthetized on ice and perfused transcardially with ice-cold 1X PBS supplemented with 10 U/mL heparin and then with 4% PFA in 1X PBS, followed by brain isolation. P0 brains were post-fixed overnight at 4°C in 4% PFA in 1X PBS. The next day the excess fixative was removed by washing the brains with 1X PBS supplemented with 0.01% (wt/vol) sodium azide (Sigma-Aldrich) for at least 2 hr at room temperature (RT).

Samples were cleared using the advanced CUBIC clearing protocol for whole-brain and whole-body clearing (*Susaki et al., 2015*). In short, samples were immersed in 1/2-water-diluted Reagent-1 containing 1 μM SYTO 16 (Thermo Fisher) and incubated at 37°C for 6 hr (Reagent-1: 25 weight% (w %) Urea, 25 wt% Quadrol, 15 wt% Triton X-100, and dH2O). 1/2 diluted Reagent-1 was replaced with Reagent-1 containing 1 μM SYTO 16 and incubated overnight at 37°C on a rotator. The next day, solution was replaced with fresh Reagent-1 containing 1 μM SYTO 16 and was incubated overnight at 37°C. Reagent-1 was replaced with fresh Reagent-1 containing 1 μM SYTO 16 every 48 hr for a total of 8 days. Tissue clearing was stopped by washing the sample with 1X PBS supplemented with 0.01% (wt/vol) sodium azide at RT once for 2 hr, once overnight and again once for 2 hr. After the wash step, samples were immersed in 10 mL of 1/2-PBS-diluted Reagent-2 (Reagent-2: 25 wt% Urea, 50 wt% sucrose, 10 wt% Triethanolamine, and dH2O). Vials containing brains in 1/2-PBS-diluted Reagent-2 were placed in a vacuum desiccator with gentle shaking overnight at RT. The following day, the solution was replaced with Reagent-2 and incubated at 37°C overnight on a rotator. The next day, Reagent-2 solution was replaced with fresh Reagent-2 and incubated at 37°C overnight on a rotator. This step was repeated four times. We imaged and processed a total of n = 15 from 5 female and 10 male mice.

## Lightsheet imaging

Samples were imaged within 7 days of completing the clearing protocol at the Gladstone Institutes Histology and Light Microscopy Core on a Zeiss Lightsheet Z.1 microscope. Lightsheet imaging of a mouse P0 brain required approximately 50 tiles (5 × 10) per brain at 5x (Zeiss EC Plan-Neofluar 5x, NA 0.16, RI 1.45, WD 5.6 mm), which afforded a level of resolution that allowed for nuclei detection (0.913 μm/px lateral resolution).

The microscope requires specimen to be suspended between fixed illuminator and detector objectives, typically using capillaries to embed small specimens. To suspend the relatively larger whole brain, we designed a custom 3D-printed rod with a platform to maximize surface area for gluing the ventral surface of the brain to the mount (*Figure 7—figure supplement 1*). The rod also contains a neck to position the brain directly under the mount holder, allowing full movement of the brain throughout the chamber to capture the brain it its entirety. The mount oriented the brain axially toward the detector to minimize the path of the illuminators on opposite sides laterally through the tissue as well as the emission path superiorly to the detector. We designed the mount in Onshape and printed it using a Stratasys uPrint 3D printer.

To glue the brain to the mount, we placed the brain, ventral surface facing up, on a custom sieve to drain reagent and rolled cotton bud applicators on the ventral surface of the brain to dry it. We applied glue (Scotch Super Glue Liquid) with a brush applicator to the mount platform surface and attached it the brain. After flipping the brain to dorsal side up, we placed the mount in a custom holder to allow the glue to dry over 3 min and dribbled Reagent-2 media on the dorsal surface to ensure that it did not dry out. After the glue dried, we suspended the mount from the microscope manipulator before immersing the mounted brain into the microscope chamber filled with Reagent-2.

With the lasers initiated, we aligned the two opposite illuminators along the z-axis by visual inspection at a central region of the brain. We ranged the z-stack from just beyond the first and last z-planes with visible nuclei, typically 800–1000 planes using a slice interval of 4.935 μm with a light-sheet thickness of 10.44 μm, and set the image tiling to include the farthest lateral and anterior-posterior nuclei with a 10% overlap per tile. We illuminated the brain with 488 nm excitation and 30 ms dwell time through the Z.1 LSFM 5x/0.1 illuminators, LBF 405/488/561/640 laser blocking filter, SBS LP 560 secondary beam splitter, and BP 505–545 band pass filter. The microscope paused for 20 s between each tile to allow tissue settling after repositioning for the next tile. We controlled the microscope through the Zeiss Zen microscopy software suite and saved images in a CZI multi-tile format with all tiles stored in a single archive.

## Image stitching

To stitch the tiled microscopy images in an automated fashion, we used the Fiji/ImageJ (*Schindelin et al., 2012*; *Rueden et al., 2017*) BigStitcher plugin (*Hörl et al., 2019*), a successor to the Stitching plugin (*Preibisch et al., 2009*) that allows for better memory management and multi-processing as well as a graphical interface to verify alignments. As we needed to stitch multiple brains, we accessed this plugin headlessly through its scriptable interface, manually intervening only to visually verify alignments before proceeding with the fusion step.

After importing the CZI file into the BigStitcher HDF5-based format, the plugin auto-detected the brightest illuminator for each tile, discarding planes from the other illuminator. We chose to simply select planes from the optimal illuminator rather than fusing planes from both illuminators after our inspection revealed that illuminators rarely if ever aligned perfectly throughout the tile, leading to artifacts such as apparent elongation of nuclei from imperfect overlap of the same nuclei from different illuminators.

The plugin calculated tile shifts using a phase correlation method, and we filtered out links below a correlation threshold of r = 0.8 before applying shifts through the two-round iterative global optimization strategy. To account for occasional tile misalignments, we manually inspected every pre-stitched brain to reposition any misaligned tiles, which occurred in approximately 10% of brains. Once tiles aligned, the plugin fused them into a single large TIFF image per channel. We imported this fused file via Python-Bioformats and Javabridge, libraries that allow access to life science formats via Bio-Formats, to a Numpy array format for image processing in our Python-based software as outlined below.

## Automated nuclei detection

We detected nuclei in cleared mouse brains using a 3D Laplacian of Gaussian blob detection technique throughout each whole brain. To perform detections in large images several hundred gigabytes (GB) to over a terabyte (TB) in size, we subdivided the image into many smaller chunks to reduce RAM requirements and maximize parallel processing. We loaded images through the Numpy library's memory mapped method (load with the mmap_mode option) to load only the necessary parts of the image on-the-fly, allowing us to load small images a chunk at a time without reading the entire volumetric image into memory. We divided the image shape into overlapping chunks to ensure that nuclei at borders would not be missed, with overlap size of approximately the nucleus diameter. After determining the offset and shape of each chunk, we set the image array as a class attribute, initiated multiprocessing (the multiprocessing.Pool in the standard Python library), and accessed each chunk as a view in a separate process via class methods to avoid duplicating arrays in memory. Thus, we could control total memory usage by the size of chunks and the number of CPU (central processing unit) cores available for separate processes.

3D cell detection poses a number of challenges including adapting to local variation such as staining inhomogeneity, background variation, and autofluorescence, in addition to overlapping cells in dense tissue (*Shuvaev et al., 2017*). To address these issues, we analyzed images in a local manner by further subdividing each chunk for preprocessing based on its immediate surroundings. We split each chunk into sub-chunks using the same approach as above but ran each sub-chunk serially within each CPU process. In each sub-chunk, we first clipped intensity values at the 5th and 98.5th percentiles (percentile in Numpy) to remove extreme outliers, rescaled the intensities from 0 to 1, and further saturated signal by clipping the rescaled intensities at the 50th percentile. We next enhanced edges by using unsharp masking with a Gaussian sigma of 8 (filters.gaussian in scikit-image) to identify sharp details as the difference between an image and its blurred version (which we amplified by a factor of 0.3) and adding back those details to the original image. We mildly eroded the resulting signal with an erosion filter using an octahedron structuring element of size 1 (morphology.erosion with morphology.octahedron in scikit-image) to separate out blobs.

To detect blobs, we implemented the 3D Laplacian of Gaussian blob detector from the scikit-image library (feature.blob_log) as a multi-scale interest point operator (*Lindeberg, 1993*). We set the minimum and maximum sigma based on the microscopy resolution, with 10 intermediate values, detection threshold of 0.1, and overlap fraction threshold of 0.55 below which duplicated blobs are eliminated. Initially we missed many nuclei positioned above one another in the z-direction, likely because the anisotropy necessitated by the relatively thick lightsheet at 5x in our setup limited

resolution in the z-direction. To improve detection along the z-axis, we interpolated the images in this direction to near isotropy before detection (transform.resize in scikit-image). The blob detector had a tendency to cluster detections in the bottom and topmost z-planes in each ROI from nuclei visible within the ROI but whose centroids are outside. To avoid this clustering and minimize duplication with adjacent ROIs, we cropped nuclei from these planes on the assumption that they would be captured better in the adjacent, overlapping ROIs.

Overlapping chunks minimized missing nuclei at edges but also necessitated pruning blobs duplicately detected in adjacent chunks. Pruning involves checking for duplicates within all potentially overlapping regions. Since the overlapping portions of the regularly spaced chunks collectively form a grid pattern throughout the full volumetric image, we could limit our search to these grid planes along each axis. After completing detection on the whole image, we first pooled all detected blobs into a single array. Along a given axis of the full image, we determined the boundaries for each overlapping region and all of its blobs. Within each overlapping region, we found all blobs close to another blob by taking the absolute value of the difference between all blobs with one another and finding blobs within a given tolerance in all dimensions. The tolerance was titrated so that the ratio of final blobs in overlapping regions to the next adjacent regions of same volume was about 1:1. For each close pair of blobs found, we replaced both blobs with a new blob that took the mean of their coordinates. To minimize memory usage, we checked smaller groups of blobs against one another until completing all comparisons. We checked overlapping regions simultaneously in multiprocessing along a given axis for efficiency, re-pooled all blobs, and pruned along the next axis to account for blobs that may have been duplicately detected in overlapping chunks along multiple axes, at grid intersections.

Parameters for preprocessing, detections, and pruning steps were optimized through a Grid Search approach, a type of hyperparameter tuning, to check combinations of parameters systematically. To evaluate the accuracy of each set of parameters, two students in our lab generated truth sets of nuclei locations and radii using our serial 2D nuclei annotation tool, taking ROIs of size 42×42×32 pixels (x, y, z) from representative ROIs of all major brain structures (n = 15 forebrain, eight midbrain, and seven hindbrain; n = 2766 nuclei). They separately generated additional truth sets at a slightly lower magnification (4x), size 60×60×14 pixels (n = 40 ROIs, 1116 nuclei) to increase representation. After detecting nuclei on these images with a given set of parameters, matches between detections and ground truth was determined using the Hungarian algorithm, a combinatorial optimization method to determine optimal assignments between two sets (*Kuhn, 1955*), as implemented in optimize.linear_sum_assignment in Scipy. After scaling nuclei coordinates for isotropy, we found the Euclidean distances between detected and truth nuclei points through distance.cdist, which serves as the cost matrix input to optimize.linear_sum_assignment to find optimal pairings between points based on closest distance. We took correctly identified detections, or true positives (TP), as pairings within a given tolerance distance. Unpaired detections or those in pairs exceeding this threshold were considered false positives (FP), and the same for ground truth were false negatives (FN). Since a match for a given nucleus within the ROI may lie outside of it and thus go unseen, we first searched for pairings only within an inner sub-ROI, followed by a secondary search for pairings between only unmatched inner sub-ROI objects and the rest of the ROI (*Ho et al., 2017*). This approach reduced the total number of nuclei available (n = 1118 nuclei) but avoided missed border matches. As measures of performance of our detection compared with ground truth, we used the following standard equations:

$$\text{Sensitivity (Recall)} = \frac{TP}{TP + FN} \tag{6}$$

$$\text{Positive Predictive Value (PPV, or Precision)} = \frac{TP}{TP + FP} \tag{7}$$

## Image downsampling and registration

To assign nuclei to the proper brain label, we employed automated label propagation by registering the E18.5 atlas to each of our imaged mouse brains using SimpleElastix (*Marstal et al., 2016*), a toolkit that combines the programmatic access of SimpleITK (*Lowekamp et al., 2013*) to the Insight Segmentation and Registration Toolkit (ITK) (*Yoo et al., 2002*) with the Elastix (*Klein et al., 2010*;

*Shamonin et al., 2014*) image registration framework. Elastix has been recently validated as a computationally efficient and accurate tool for registration of mouse brains cleared by CUBIC (*Nazib et al., 2018*).

As stitched images are typically several hundreds of GBs per file, image downsampling was necessary to reduce memory utilization during image registration. To reduce file size efficiently in both time and memory usage, we employed the same chunking strategy as used during nuclei detection except with larger, non-overlapping units to resize multiple sections of the image simultaneously. We also reduced memory required for the output array by saving directly to disk with a memory-mapped array (lib.format.open_memmap in Numpy). We matched the target final size to the E18.5 atlas, which would be registered to each downsampled image.

Registration involved rigid followed by non-rigid alignment. For rigid registration, we employed a translation (translation parameter map in SimpleElastix) with default settings except increasing to 2048 iterations (MaximumNumberOfIterations setting) followed by an affine (affine parameter map) with 1024 iterations, applied with an ElastixImageFilter, to shift, resize, and shear the atlas microscopy image to the same space as that of the sample brain. For non-rigid alignment, we employed a b-spline strategy (bspline parameter map) guided by the AdvancedNormalizedCorrelation similarity metric (*Nazib et al., 2018*; *Hammelrath et al., 2016*) with grid spacing of size 60, measured in voxels rather than physical units (FinalGridSpacingInVoxels setting in place of FinalGridSpacingInPhysicalUnits), over 512 iterations. The TransformixImageFilter in SimpleElastix allowed us to apply the identical registration transformation to the atlas labels image, except that we set the final b-spline interpolation order (FinalBSplineInterpolationOrder) to 0 to avoid interpolating any new values, preserving the labels' specific set of integer values. We applied this identical transformation to both the mirrored and edge-refined atlas labels.

To evaluate the level of alignment from registration, we measured the similarity between each registered atlas histology and its corresponding sample image using a Dice Similarity Coefficient (DSC) (*Dice, 1945*) as implemented by the GetDiceCoefficient function in SimpleElastix/SimpleITK, given by the equation (*Tustison and Gee, 2009*):

$$\mathrm{Dice\,Similarity\,Coefficient\,(DSC)} = 2\frac{|S \cap T|}{|S| + |T|} \tag{8}$$

where S and T are two different sets of voxels. We took the foreground of each atlas and sample microscopy image to be its mean threshold (filters.threshold_mean function in scikit-image) and input them to a LabelOverlapMeasuresImageFilter to take the DSC.

## Whole-brain nuclei measurements by label

Registration of the atlas to our volumetric nuclear-stained brain microscopy images allowed us to quantify nuclei per label for comparison with the original (mirrored) and smoothed (edge-aware) atlases. We first measured volumes per label by taking a mask of each registered label and summing the foreground pixels within each mask before multiplying this volume by the microscopy pixel resolution (scaled for downsampling) to obtain volumes in physical units (mm$^3$).

For nuclei densities, we first constructed a nuclei heat map by converting the nuclei coordinates to nuclei per voxel within the downsampled image. We scaled the coordinates to the scaling of the downsampled image, rounding to the nearest integer, and found the counts of nuclei at each coordinate (unique with return_counts option in Numpy). We next indexed these coordinates directly into an empty Numpy array of the same shape as that of the downsampled image to assign them to the corresponding nuclei counts at each voxel. We used the same label mask previously obtained to find the number of nuclei within each given label. Dividing the number of nuclei by the volume within each label gave the label nuclei density.

To measure the variability of nuclei within each label before and after label reannotation, we measured the coefficient of variation within each label given by the standard equation:

$$\mathrm{Coefficient\,of\,variation\,(CV)} = \frac{\sigma}{\mu} \tag{9}$$

where $\sigma$ is the standard deviation, and $\mu$ is the mean. A lower coefficient of variation indicates lower variability and thus tighter capture of a more homogeneous label. As a raw proxy for nuclei variation,

we first measured the variation of intensities within the nuclear-stained lightsheet images. Using the same label masks, we took the standard deviation of voxel intensities and divided it by the mean of intensities within the label (std and mean, respectively, in Numpy) to obtain the intensity coefficient of variation. Similarly, we measured the nuclei coefficient of variation by measuring the standard deviation and mean values of nuclei counts per label within the nuclei heat map. We also stratified nuclei ROIs into low/medium and high-density regions by taking the Otsu threshold of their density, which corresponded with their density histogram, and separately measured recall and precision in each group (*Figure 6—figure supplement 2D*).

## Nuclei clustering

As another measure of label alignment at the nuclei level, we measured nuclei clustering using Density-Based Spatial Clustering of Applications with Noise (DBSCAN) (*Ester et al., 1996*) as implemented by the scikit-learn library (*Pedregosa et al., 2018*). DBSCAN clusters tightly packed points within a neighbor distance defined by the parameter $E$ and a minimum number of points given as another parameter, with isolated points in lower density regions that cannot be clustered considered 'outliers' or 'noise.' The minimum number of samples is typically taken as $2 \cdot ndim$, where $ndim$ is the number of dimensions (*Schubert et al., 2017*), thus giving 6 for our 3D nuclei point cloud. To find $E$, the nearest-neighbor distance of the $2 \cdot ndim - 1$ neighbor for each point is sorted and plotted to find the distance at the 'elbow' point, the point of maximum curvature (*Schubert et al., 2017*), which we found to be at least 20 µm (*Figure 7—figure supplement 3A*).

For each label in the original (mirrored) atlas registered to each wild-type brain, we extracted the nuclei coordinates within the label and clustered them by DBSCAN to find the number of clusters, nuclei per cluster, and 'noise,' or number of isolated nuclei that remained unclustered. We repeated the same process using the same nuclei coordinates for each brain but with the smoothed (edge-aware) atlas registered identically to the brain. As the 'elbow' distance of maximum curvature can be difficult to define, and $E$ can strongly influence the clustering, we repeated this process for a range of $E$ values through the elbow region (*Figure 7—figure supplement 3B*) and highlighted a conservative distance of 20 µm (*Figure 7—figure supplement 3C–E*).

## Statistics

Statistical tests used for comparisons are described in the relevant Results sections and figure legends. Bonferroni correction for p-values is applied when multiple statistical tests are performed for each figure sub-panel as indicated for the number of tests.

## MagellanMapper software suite

We provide the MagellanMapper image software suite as a tool to assist with visualization, annotation, and automated processing of volumetric images (*Figure 7—figure supplement 4*). The suite consists of a graphical user interface (GUI) to aid visualization of 2D images in a 3D context and command-line interface (CLI) for non-interactive processing in workstation and cloud environments.

## Graphical interface

The main GUI integrates ROI selection with 3D point and surface rendering through the Mayavi toolkit (*Ramachandran and Varoquaux, 2011*). Users can load volumetric image files and specify ROI boundaries through sliders and text boxes or load a previously saved ROI. 3D point rendering provides a voxel-based visualization of the ROI with minimal filtering, whereas the 3D surface rendering utilizes VTK (The Visualization Toolkit) (*Schroeder et al., 1998*) for cleaner images. The interface is implemented in TraitsUI for integration with Mayavi.

To inspect raw images, the user can launch two types of mixed 2D/3D interfaces from the main GUI to display and annotate the original 2D images for each plane. The first 2D interface is a serial 2D ROI viewer that shows each successive 2D plane within the ROI side-by-side, allowing the user to follow objects such as nuclei that come and go from plane to plane. Larger overview images at different magnifications show context and synchronize with the smaller views to effectively zoom in on a given plane. These overview images are also scrollable along the z-axis to visualize subtle object shifts in place. The interface also provides annotation tools geared toward blob detection such as nuclei. Detected blobs appear as circles along with optional segmentations, and the user can drag,

resize, or cut/copy/paste circles to improve placement and flag their correctness. We have used this simplified blob annotator to generate truth sets for blob detection verification and optimization.

In addition to ROI viewers, we provide a simultaneous orthogonal viewer to visualize and annotate atlases in all three dimensions. It displays orthogonal planes of the full volumetric image in three separate panels, with crosshairs in each panel denoting the corresponding planes in other panels. Clicking on or scrolling within any panel updates crosshairs and synchronizes the other displayed planes. The user can also load label maps to overlay directly on atlas microscopy images with an adjustable labels opacity to allow close inspection of annotation alignment with anatomical structures. To distinguish an arbitrary number of labels from one another, we use a custom discrete colormap with randomly generated colors, each assigned to a single label. We display all images as views of one another to minimize memory requirement and update all orthogonal planes in real-time.

As a method for simple, rapid editing of these labels, the user can enter an editing mode to simply click and drag on individual labels to paint them into other spaces. We have used the interface with a tablet and electronic pen to edit labels by drawing. As hand-editing any given plane likely introduces edge artifacts seen in other orthogonal directions, we also designed a method to interpolate contours between two distant planes. After the user edits the same label ID at start and ending planes and initiates the interpolation, it takes the signed distance transforms of the label masks (ndimage.distance_transform_edt in Scipy combined with a mask of the original label to identify distances inside versus outside the label) and interpolates those distances for each intervening plane (interpolate.interpn applied across a mesh-grid of the planes). The interpolation provides label border extensions that are generally smooth in all dimensions after manually editing only the first and last plane.

The annotation interfaces are implemented in Matplotlib (*Hunter, 2007*). Blobs are stored in an SQLite (*Hipp et al., 2021*) database, while atlas edits are saved directly to their underlying 3D image file.

## Headless pipelines

In addition to a GUI for interactive visualization and verification, MagellanMapper provides automated pipelines for non-interactive image processing such as whole brain nuclei detection in cloud-based work environments. Users can access the suite via its CLI, and the suite provides Bash scripts to connect MagellanMapper with other tools such as Fiji/ImageJ for image stitching and Amazon Web Services (AWS) in a platform-independent manner.

For input/output (I/O), the suite utilizes standard 3D image formats for portability with other software libraries. Microscopy images stored in proprietary formats such as Zeiss CZI format can be imported to a standard Numpy array archive using the Bio-Formats library (*Linkert et al., 2010*) (via the Javabridge and Python- Bioformats libraries developed for CellProfiler [*Lamprecht et al., 2007*] libraries) along with a separate Numpy archive containing image metadata extracted from the original file. Loading the imported Numpy array as a memory-mapped file (load function with mmap_-mode option) allows users to access small parts of large files to minimize load time and memory usage by only loading the requested ROI rather than the potentially TB-sized full volumetric image. In addition to Numpy array archives, other 3D image formats such as MetaImage, NIfTI, and DICOM are supported through the SimpleITK/SimpleElastix library. Annotated images are saved to their respective formats.

As an example of an automated pipeline for image import, a user can launch the script from a cloud-based server instance to first retrieve and decompress an archived microscopy file previously saved in a cloud storage location. After extracting the microscopy image, the pipeline script launches Fiji/ImageJ to run a custom headless script for the BigStitcher plugin, which stitches the tiled microscopy image as described above. After allowing the user to verify and adjust tile placements through the BigStitcher interface, the script continues the BigStitcher tile fusion operation, resulting in TIFF formatted images for each channel. The pipeline script next calls MegellanMapper to import these TIFF images into a single Numpy array archive. Subsequent pipelines can be run to process the imported image for automated nuclei detections, downsample or transpose the image, register images to an atlas, or automatically refine new atlases in 3D.

## Software access

We provide MagellanMapper as open-source software in the hope of facilitating both interactive and headless processing of large volume microscopy images and the atlases to which they will be registered.

The suite API includes 3D image processing functions designed to be useful as library methods for other applications. Library functions for the Python and R plots depicted here are also provided to reproduce similar graphs. As an open-source, Python-based tool, our vision for the suite is that it will work alongside, integrate with, and itself become refined by the many other excellent image processing software suites and libraries available to the scientific community.

## Software versions

The MagellanMapper suite is written in Python with the following versions and Python libraries: Python 3.6, Numpy 1.15, Scipy 1.1, scikit-image 0.14, scikit-learn 0.21, Matplotlib 3.0, Matplotlib ScaleBar 0.6, Mayavi 4.6, TraitsUI 6.0, PyQt 5.11, Python-Bioformats 1.1, Javabridge 1.0, SimpleElastix 1.1, and Pandas 0.23. We used Conda 4.7 for Python library management. We stitched images with Fiji/ImageJ 1.52 using BigStitcher 0.2.10. We computed additional statistics in R 3.5 with RStudio 1.1. To interact with AWS, we used AWS CLI 1.16 and Boto3 1.9. For image acquisition, we used Zeiss Zen 2014.

We conducted most software development on a Mid-2014 MacBook Pro (Intel Core i7-4980HQ, 16 GB RAM, 1TB SSD) with MacOS 10.13, image stitching and nuclei detections on AWS EC2 c5.9xlarge instances (36 vCPUs, 72 GB RAM, typically configured with 1TB SSD) running RHEL 7.5 and Ubuntu 18.04, and volume measurements and aggregation on a Dell Precision T7500 workstation (64 GB RAM, 256 GB SSD and 2.2TB HDD) with Ubuntu 18.04. We conducted additional cross-platform compatibility testing on a Microsoft Surface Pro (2017) laptop (8 GB RAM, 256 GB SSD) with Microsoft windows 10 (build 1803), Windows Subsystem for Linux running Ubuntu 18.04, and Windows 10 virtual machines in VirtualBox 6.0.

## Acknowledgements

We would like to thank the reviewers for their constructive feedback, including Harold Burgess. We are grateful for the microscopy advice and expertise from Meredith Calvert, director of the Gladstone Institutes Histology and Light Microscopy Core. We greatly appreciate Eric Lam from the KAVLI-PBBR Fabrication and Design Center at UCSF for assistance with the initial design of the custom 3D printed microscopy brain mount. We are deeply indebted to Hwee Kuan Lee from the Agency for Science, Technology and Research (A*STAR) for invaluable imaging informatics advice. We appreciate support from the Institute of Molecular and Cell Biology at A*STAR.

## Additional information

### Funding

| Funder | Grant reference number | Author |
|---|---|---|
| Brain and Behavior Research Foundation | NARSAD Young Investigator Grant | Stephan J Sanders |
| National Institute of Mental Health | U01 MH122681 | Stephan J Sanders |
| National Institute of Mental Health | R01 MH109901 | Stephan J Sanders |
| National Institute of Neurological Disorders and Stroke | R01 NS099099 | John LR Rubenstein |

The funders had no role in study design, data collection and interpretation, or the decision to submit the work for publication.

## Author contributions
David M Young, Data curation, Software, Formal analysis, Validation, Investigation, Visualization, Methodology, Writing - original draft, Writing - review and editing, Designed all experiments, developed the 3D reconstruction tool, performed and optimized imaging; Siavash Fazel Darbandi, Data curation, Investigation, Methodology, Performed mouse husbandry and tissue clearing, assisted with imaging; Grace Schwartz, Investigation, Optimized tissue clearing and sample mounting; Zachary Bonzell, Methodology, Piloted tissue clearing and imaging; Deniz Yuruk, Mai Nojima, Data curation, Validation, Annotated ground truth sets, software usability testing; Laurent C Gole, Methodology, Computer vision and image processing guidance, figure review; John LR Rubenstein, Resources, Supervision, Provided expertise on mouse development and anatomy; Weimiao Yu, Supervision, Methodology, Writing - review and editing; Stephan J Sanders, Conceptualization, Resources, Data curation, Formal analysis, Supervision, Funding acquisition, Investigation, Visualization, Methodology, Writing - original draft, Project administration, Writing - review and editing, Supervised the entire project

## Author ORCIDs
David M Young ⓘD https://orcid.org/0000-0003-3303-2001
John LR Rubenstein ⓘD http://orcid.org/0000-0002-4414-7667
Stephan J Sanders ⓘD https://orcid.org/0000-0001-9112-5148

## Ethics
Animal experimentation: All procedures and animal care were approved and performed in accordance with institutional guidelines from the University of California, San Francisco Laboratory Animal Research Center (LARC). All animal handling complied with the approved Institutional Animal Care and Use Committee (IACUC) protocol (AN180174-02) at the University of California, San Francisco.

## Decision letter and Author response
Decision letter https://doi.org/10.7554/eLife.61408.sa1
Author response https://doi.org/10.7554/eLife.61408.sa2

# Additional files

## Supplementary files
• Supplementary file 1. Spreadsheet mapping colors to label names and IDs in the Allen ontology for each atlas in the ADMBA.

• Supplementary file 2. Spreadsheet of metric differences by label for WT brains before and after edge-aware atlas refinement.

• Transparent reporting form

## Data availability
The full 3D generated atlases and wild-type brain images are being deposited with the Human Brain Project EBRAINS data platform. All data analyses are included in the manuscript and supporting files.

The following datasets were generated:

| Author(s) | Year | Dataset title | Dataset URL | Database and Identifier |
|---|---|---|---|---|
| Young D, Yu W, Sanders SJ | 2020 | E11.5 3D Edge-Aware Refined Atlas Derived from the Allen Developing Mouse Brain Atlas | https://doi.org/10.25493/H9A3-GFT | EBRAINS , 10.25493/H9A3-GFT |
| Young D, Yu W, Sanders SJ | 2020 | E13.5 3D Edge-Aware Refined Atlas Derived from the Allen Developing Mouse Brain Atlas | https://doi.org/10.25493/YP9K-YMW | EBRAINS , 10.25493/YP9K-YMW |
| Young D, Yu W, Sanders SJ | 2020 | E15.5 3D Edge-Aware Refined Atlas Derived from the Allen Developing | https://doi.org/10.25493/EXET-XND | EBRAINS , 10.25493/EXET-XND |

| | | | | | |
|---|---|---|---|---|---|
| | | | Mouse Brain Atlas | | |
| Young D, Yu W, Sanders SJ | 2020 | E18.5 3D Edge-Aware Refined Atlas Derived from the Allen Developing Mouse Brain Atlas | https://doi.org/10.25493/X4ZT-ARE | EBRAINS , 10.25493/X4ZT-ARE |
| Young D, Yu W, Sanders SJ | 2020 | P4 3D Edge-Aware Refined Atlas Derived from the Allen Developing Mouse Brain Atlas | https://doi.org/10.25493/QYP4-5VQ | EBRAINS , 10.25493/QYP4-5VQ |
| Young D, Yu W, Sanders SJ | 2020 | P28 3D Edge-Aware Refined Atlas Derived from the Allen Developing Mouse Brain Atlas | https://doi.org/10.25493/YW1E-6BW | EBRAINS , 10.25493/YW1E-6BW |
| Young D, Yu W, Sanders SJ | 2020 | P56 3D Edge-Aware Refined Atlas Derived from the Allen Developing Mouse Brain Atlas | https://doi.org/10.25493/MYPD-QB8 | EBRAINS , 10.25493/MYPD-QB8 |

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
