## [Decision Letter]

**Acceptance summary:**

The paper provides MagellanMapper, a suite of tools to provide quantitative measures of brain structure to accurately labeled 3D digital atlases across mouse neural development, and demonstrates that the resulting brain parcellations are superior to a naive agglomeration of the existing 2D labels. The novel computational methods transform slice annotations in the Allen Developing Mouse Brain Atlas into digital 3D reference atlases. The response to reviewer comments was complete and persuasive.

**Decision letter after peer review:**

Thank you for submitting your article "Constructing and Optimizing 3D Atlases From 2D Data With Application to the Developing Mouse Brain" for consideration by *eLife*. Your article has been reviewed by two peer reviewers, and the evaluation has been overseen by Kate Wassum as the Senior Editor and Joseph Gleeson as Reviewing Editor. The following individual involved in review of your submission has agreed to reveal their identity: Harold Burgess (Reviewer #3).

The reviewers have discussed the reviews with one another and the Reviewing Editor has drafted this decision to help you prepare a revised submission.

We would like to draw your attention to changes in our revision policy that we have made in response to COVID-19 (https://elifesciences.org/articles/57162). Specifically, when editors judge that a submitted work as a whole belongs in *eLife* but that some conclusions require additional new data, as they do with your paper, we are asking that the manuscript be revised according to the guidelines below as much as possible, or for items that cannot be directly addressed due to COVID-19, either limit claims to those supported by data in hand, or to explicitly state that the relevant conclusions require additional supporting data. Our expectation is that the authors will eventually carry out the additional experiments and report on how they affect the relevant conclusions either in a preprint on bioRxiv or medRxiv, or if appropriate, as a Research Advance in *eLife*, either of which would be linked to the original paper.

Despite the availability of a high resolution, expertly annotated digital adult mouse brain atlas (Allen CCFv3), accurately labeled 3D digital atlases across mouse neural development are lacking. The authors have filled that gap by developing novel computational methods that transform slice annotations in the Allen Developing Mouse Brain Atlas into digital 3D reference atlases. They demonstrate that the resulting brain parcellations are superior to a naive agglomeration of the existing 2D labels, and provide MagellanMapper, a suite of tools to aid quantitative measures of brain structure. Cellular level whole-brain quantitative analysis is rapidly becoming a reality in many species and this manuscript provides a foundational resource for mouse developmental studies. The methods are sophisticated, carefully applied and thoroughly evaluated. The manuscript reports a computational approach to transforming available 2D atlases of mouse brains into the 3D volumetric datasets. By optimizing the "smoothing" steps, a better quality of such 3D atlases is produced claimed. In addition, the authors applied their method to the imaging dataset of neonatal mouse brains obtained by lightsheet microscopy, as proof of its potential utilization in research.

1) The pipeline of the method involved the "mirroring" before the "smoothing" steps. Is it possible to perform the "smoothing" of one hemisphere and then "mirror" the smoothed 3D atlas onto the other hemisphere to check for the alignment? By doing so, the other hemisphere could serve as an internal control for the quality and accuracy of the 3D atlas.

2) The authors developed the “edge-aware procedure”, employed to extend existing labels to unannotated lateral regions of the brain, taking advantage of intensity gradations in underlying microscope images. Authors should manually annotate a small part of the lateral brain region to compare accuracy and compare computationally generated labels to the partial lateral labels in P28 brain.

3) For more delicate subregions (e.g., those in the hypothalamus) without clear anatomical boundaries, this “edge-aware” adjustment step may become ineffective. What could then be done for these subregions? Also, it is important to note that the anatomical edges required the manual annotation.

4) Annotations present in the ADMBA took advantage of co-aligned ISH data (and computational approaches using co-aligned gene expression data have been used for de novo brain parcellation). Intensity differences in the light-microscope images may not provide enough contrast or access this expression data for for accurate segmentation. Could there be instances where adjacent regions do not have intensity differences, and the edge-aware procedure actually reduces the accuracy of the manual annotation? What is the evidence that contrast is sufficient to demark the boundaries?

5) It does appear that despite the care to avoid losing thin structures, there is some loss, for example for the light-green structure in the forebrain in Figure 5E. Authors should indicate if all labels were preserved, and provide information on volume changes by label size.

6) The accuracy of non-rigid registration of light-sheet images to the references is assessed only using a DSC value for whole-brain overlaps. This does not assess the precision of registration within the brain. The authors should apply some other measure to assess quality of alignment within the brain (e.g. mark internal landmarks visible in the reference and original light-sheet images, and measure the post-registration distance between them).

---

## [Author Response]

Despite the availability of a high resolution, expertly annotated digital adult mouse brain atlas (Allen CCFv3), accurately labeled 3D digital atlases across mouse neural development are lacking. The authors have filled that gap by developing novel computational methods that transform slice annotations in the Allen Developing Mouse Brain Atlas into digital 3D reference atlases. They demonstrate that the resulting brain parcellations are superior to a naive agglomeration of the existing 2D labels, and provide MagellanMapper, a suite of tools to aid quantitative measures of brain structure. Cellular level whole-brain quantitative analysis is rapidly becoming a reality in many species and this manuscript provides a foundational resource for mouse developmental studies. The methods are sophisticated, carefully applied and thoroughly evaluated. The manuscript reports a computational approach to transforming available 2D atlases of mouse brains into the 3D volumetric datasets. By optimizing the "smoothing" steps, a better quality of such 3D atlases is produced claimed. In addition, the authors applied their method to the imaging dataset of neonatal mouse brains obtained by lightsheet microscopy, as proof of its potential utilization in research.1) The pipeline of the method involved the "mirroring" before the "smoothing" steps. Is it possible to perform the "smoothing" of one hemisphere and then "mirror" the smoothed 3D atlas onto the other hemisphere to check for the alignment? By doing so, the other hemisphere could serve as an internal control for the quality and accuracy of the 3D atlas.

Thank you for suggesting the use of one hemisphere as an internal control for the other hemisphere. We’ve added these analyses as Figure 4—figure supplement 3 and we reference the analysis in the main text. In our current setup, we mirror both the labels and the underlying histological image to the opposite hemisphere since the histological images have asymmetries that would not correspond with the mirrored labels. Because the two hemispheres are now identical, smoothing before versus after mirroring would also be identical. Alternatively, we could mirror the labels only (i.e. not the underlying histological image) to the opposite hemisphere before smoothing. However, the "edge-aware" smoothing step is based on the assumption that the core of each label obtained through erosion represents pixels with the highest probability of accurate placement, and for many labels this assumption is not correct for the other hemisphere due to the asymmetry.

However, we can use the asymmetric hemispheres as internal controls if we register the mirrored atlases back to their unmirrored versions. This registration allows us to compare the quality and accuracy in the hemisphere on which the atlases were not derived, tested through segmentation by 3D image registration, which is a typical use case of the atlases. We can then compare the quality and accuracy between the unsmoothed and smoothed labels.

Taking each of our original (mirrored) and smoothed (edge-aware) atlases across the ADMBA, we registered each atlas to its original (unmirrored) histological image (Figure 4—figure supplement 3A). We first tested the hemisphere from which the atlases were derived and observed a significant improvement with the smoothed atlas for within region homogeneity (median coefficient of variation 0.340 unsmoothed vs. 0.331 smoothed, p = 0.008, WSRT; Figure 4—figure supplement 3B) and distance between anatomical and label edges (median of 91 million µm unsmoothed vs. 50 million µm smoothed, p = 0.008, WSRT, Figure 4—figure supplement 2C). Repeating this on the contralateral, asymmetric hemispheres as an internal control, we observed similar evidence of improvement for both within region homogeneity (median of 0.298 unsmoothed vs. 0.291 smoothed, p = 0.008, WSRT; Figure 4—figure supplement 3D) and distance between anatomical and label edges (median of 89 million µm unsmoothed vs. 41 million µm smoothed, p = 0.008, WSRT; Figure 4—figure supplement 3E). By testing the generalizability of the 3D reconstructed atlases to new hemispheres through this internal control, we demonstrate that we can obtain similar improvements in anatomical fidelity when applied to other brains through image registration.

2) The authors developed the “edge-aware procedure”, employed to extend existing labels to unannotated lateral regions of the brain, taking advantage of intensity gradations in underlying microscope images. Authors should manually annotate a small part of the lateral brain region to compare accuracy and compare computationally generated labels to the partial lateral labels in P28 brain.

This is a great idea and one that we had not considered before. Again, we’ve added these analyses as Figure 4—figure supplement 4, and we reference the analysis in the main text. To measure the accuracy of the computationally generated labels, we manually annotated structures in the P28 lateral planes, which also allowed comparison with the partial lateral labels in the original P28 atlas. We annotated the hippocampus and Layer 1 as the most clearly demarcated structures in these planes (Figure 4—figure supplement 4D) and compared them against the equivalent composite of sub-structures in the original, partially annotated planes (Figure 4—figure supplement 4A-C) and the 3D reconstructed labels (Figure 4—figure supplement 4E-G) using the Dice Similarity Coefficient (DSC) metric. The DSC for the original labels was 0.65 (weighted average by volume from a DSC of 0.81 for the hippocampus and 0.54 for Layer 1), compared with 0.75 for the computationally generated labels (0.82 for the hippocampus and 0.71 for Layer 1), supporting the accuracy of our imputed labels. Of note, the partially labeled planes are not used in the computationally generated planes, which essentially start from scratch using the last fully labeled plane while still yielding a higher DSC than for the partial labels and the artifact reduction evident in the axial and coronal views (Figure 4—figure supplement 4A-B, F-G).

To account for the missing labels in the original planes, we masked the ground truth to the extent of all labels in each plane before calculating the DSC, which favors the original, partial labels but avoids penalizing them where they were simply unannotated. Without this masking, the original labels' DSC drops to 0.42 (hippocampus: 0.50, Layer 1: 0.37), compared with an essentially unchanged DSC of 0.74 (hippocampus: 0.82, Layer 1: 0.69) in the computationally generated labels. We also compared the partially labeled planes with the generated planes, which showed a weighted average DSC of 0.58 across all labels, or 0.70 for the same structures compared in the manual annotations (hippocampus: 0.79, Layer 1: 0.63), consistent with the discrepancy observed between each set of labels' similarity with the manual annotations and the fact that these partially labeled planes are not used in the computational label generation.

3) For more delicate subregions (e.g., those in the hypothalamus) without clear anatomical boundaries, this “edge-aware” adjustment step may become ineffective. What could then be done for these subregions? Also, it is important to note that the anatomical edges required the manual annotation.

At the heart of this issue is the challenge that manual annotation is likely superior in the sagittal plane, but leads to substantial artifacts in the axial and coronal planes. We have demonstrated that our procedure improves the axial and coronal views, however this is bound to be at the expense of the accuracy of the sagittal alignment. As the reviewers suggest, the key question becomes, whether this trade off is worthwhile, especially in these delicate subregions. We have demonstrated that the answer is “yes” qualitatively and quantitatively across the brain, and at the reviewers’ suggestion, we now also focus on two complicated subregions in greater depth. Again, we are confident the answer is “yes”, based on the minimal changes in the boundary location in the sagittal plane in the absence of clear anatomical boundaries (see Figure 4—figure supplement 5A, and also Author response image 1 and Figure 4—figure supplement 5B in point 4). In short, if anatomical data does not support moving the boundary, we designed our approach to smooth the boundary in the axial and coronal planes without moving the location in any of the planes.

The edge-aware step performs two major functions: 1) erosion of each label to its core, the annotated pixels of highest confidence, which removes the jagged border artifact, and 2) regrowing this high-confidence core to meet either anatomical boundaries if they are present, or if not, simply to meet neighboring labels in the middle where the original boundary was located. This approach allows labels to take advantage of smoothing by the edge-aware step even without nearby anatomical boundaries while maintaining their core location, shape, and spatial relationships. Because some edge artifacts persist after this step, we further smooth each label through the morphological filter step, which smooths each label in-place without using anatomical information.

To illustrate this approach in the hypothalamus, a delicate subregion not known for its clear demarcations, we show the labeled ventromedial hypothalamic nucleus (VMH; Figure 4—figure supplement 5A, top row), which contains a combination of labels near clear boundaries and those that are not (Figure 4—figure supplement 5A, upper middle row). The anatomical surface map, shown here as edges in 2D, demarcates these clear boundaries while more ambiguous boundaries remain undemarcated (Figure 4—figure supplement 5A, lower middle row). After the erosion step reduced each label to its high-confidence core, these eroded labels are each bounded on three sides by anatomical boundaries, but no boundary exists between them. During the watershed step, these labels' regrowth is guided by the anatomical boundaries on these three sides but simply meet in the middle of the two labels, where the original boundary was located, in the absence of a clear anatomical boundary. The subsequent morphological smoothing step further reduced remaining artifacts (Figure 4—figure supplement 5A, bottom row). The coronal and axial views reveal the impact of smoothing even in the absence of anatomical boundaries by removal of the jagged edge artifact while respecting the core position of each label.

4) Annotations present in the ADMBA took advantage of co-aligned ISH data (and computational approaches using co-aligned gene expression data have been used for de novo brain parcellation). Intensity differences in the light-microscope images may not provide enough contrast or access this expression data for for accurate segmentation. Could there be instances where adjacent regions do not have intensity differences, and the edge-aware procedure actually reduces the accuracy of the manual annotation? What is the evidence that contrast is sufficient to demark the boundaries?

This question builds on the concepts raised in point 3, since our simplest answer to this is that we designed the procedure to only move a boundary in the presence of clear anatomical data. While we did not utilize the ISH data directly, by preserving the original boundaries of most labels we maintain this information in the final atlas. We are very interested in integrating gene expression data into atlas boundary identification in a systematic manner and hope to pursue this direction in our future work in this field either by integrating existing ISH data or with emerging spatial transcriptomic approaches.

We are not aware of a record of when, where, and which markers in the ISH data were used to define boundaries in the initial atlas. However, from the ADMBA reference paper (Thompson CL, et al., 2014), we can see that *Wnt3a* was used to define boundaries as a ligand "selectively expressed in the E13.5 cortical hem," where the ISH signal is contained within the larger cortical hem region. Therefore, we examined this region in a similar manner to the ventromedial hypothalamus (Figure 4—figure supplement 5A) and show that the region expressing *Wnt3a* (Author response image 1, dashed circle) remains contained within similar boundaries after atlas smoothing. The paper also presents *Hoxa2* as a marker expressed predominantly in the hindbrain during early embryonic periods up through E15.5, except only minimally in the prepontine hindbrain, the region just superior to the pontine hindbrain. We examined the hindbrain in the E15.5 atlas and show preservation of the boundary between the prepontine hindbrain and the pontine hindbrain before and after smoothing, matching the boundary shown by the *Hoxa2* ISH signal (Author response image 1, dashed line; PPH = prepontine hindbrain; PH = pontine hindbrain).

**Author response image 1. sa2fig1:** Preservation of regions demarcated by ISH markers. (A) The Wnt3a ISH signal (dashed circle) shown in Thompson CL, et al. (2014, Figure 2; top row) is expressed selectively within the cortical hem in the original labels of the E13.5 atlas (lower middle row, orange structure) and remains contained in this region in the 3D reconstructed atlas (bottom row). (B) The Hoxa2 ISH signal demarcates the border between the pontine hindbrain and prepontine hindbrain in the E15.5 atlas (top row). This boundary remains well-demarcated before (lower middle row) and after (bottom row) 3D reconstruction.

We acknowledge that some adjacent regions do not have intensity differences to demarcate their boundaries. To minimize the chance of reducing the accuracy of the original manual annotation in such regions, we have used a relatively coarse anatomical edge map (generated as a surface map in 3D) to capture only clearly demarcated boundaries with strong contrast differential. Similarly to the ventromedial hypothalamus (Figure 4—figure supplement 5A), we further illustrate these boundaries in the pontine hindbrain examined (Author response image 1), an area of low contrast around many labels but also a few more clearly delineated anatomical boundaries near other labels (Figure 4—figure supplement 5B, upper rows). By using a large σ in the Laplacian of Gaussian boundary detection on the microscopy images, we limited the number of artifactual boundaries around low contrast regions. By performing the boundary detection in 3D, we also reduced detection of contrast differences limited to a single plane.

The resulting anatomical map demarcated only clear, gross boundaries for the edge-aware step, with a minimum of artifactual boundaries that might otherwise reduce the accuracy of the original labels (Figure 4—figure supplement 5B, lower middle row). Labels nearby these boundaries are influenced by and thus take advantage of them, whereas labels far away from boundaries are predominantly influenced by neighboring labels instead, meeting between them and retaining their overall original position and shape (Figure 4—figure supplement 5B, bottom row). In this way, we derive the sufficiency to demarcate boundaries from the original labels, reshaping them only when a label's boundary is nearby but not completely aligned with a clear anatomical boundary. In general, we qualitatively noticed larger artifacts in the original labels, such as the jaggedness in the orthogonal planes, than artifacts caused by the edge-aware step, which is designed to improve accuracy while respecting and maintaining the overall shape of the manual annotations. We have clarified this description in the manuscript text.

5) It does appear that despite the care to avoid losing thin structures, there is some loss, for example for the light-green structure in the forebrain in Figure 5E. Authors should indicate if all labels were preserved, and provide information on volume changes by label size.

Thank you for pointing out this omission in the text. Investigating this led to further improvements in the atlas. All labels in the original hemisphere were preserved across all of the atlases and we now state this in the text. For completeness, we note that labels positioned solely across the sagittal midline on the otherwise unlabeled hemisphere were cut off to preserve the placement of the rest of the labels and the symmetry of the brain without duplicating the underlying microscopy image at the midline as indicated in the manuscript (Materials and methods, "Atlas 3D rotation and mirroring,").

Instead of the complete loss of a structure, the loss of the light-green structure that the reviewer notes represents variation in how far a thin appendage of a larger structure (medial pallium in the P4 atlas) extends. We have examined this region in detail (see Author response image 2) and find the cause to be due to inconsistencies in the original atlas labels.

As seen in the sequence of sagittal planes, two of six planes are missing annotations in this label, visible as two separate gaps in the axial views (Author response image 2, 152 and 155). Of the remaining four labeled planes, three are missing connections to the main structure, instead stopping at a harsh border (Author response image 2, 151, 154, 156). Only panel 153 shows a consistent and connected appendage.

**Author response image 2. sa2fig2:** Interpolation to fill gaps in and recover a thin substructure. (A) A thin extension in the medial pallium (green structure) in the P4 atlas is missing sagittal annotations in planes 152 and 155, visible as two gaps in the axial and coronal view, as well as gaps between the extension and the rest of the structure in planes 151, 154, and 155. (B) Reconnecting this extension (left) and interpolating between the edited planes fills the gaps as seen in the axial view (middle), recovering this thin substructure in the final smoothed atlas. (C) This substructure is apparent in the anterior section of the original atlas (left) but lost in the 3D reconstructed atlas (middle), although the rest of the medial pallium is present in other planes. After interpolation and 3D reconstruction, the substructure is recovered (right).

To test the extent to which these inconsistencies in the underlying atlas led to the loss of this thin structure, we manually edited the first and last plane to connect it to the main structure (Author response image 2, left), followed by edge interpolation between these planes to fill in the missing planes (Author response image 2, center). After the edge-aware step with a slightly reduced filter kernel size (8 to 7) and final smoothing step on the entire atlas, the thin structure was maintained, and harsh edges adapted to the underlying contours in the microscopy image (Author response image 2, right). Compared with the planes in Figure 5E, the structure is retained and refined with this interpolation (Author response image 2).

While this interpolation procedure can be used to rescue this structure, we are hesitant to apply such methods across the entire atlas since it is hard to distinguish such inconsistencies from the artifacts we are trying to remove. An alternative approach to rescuing this structure would be to reduce the maximum erosion filter kernel size, but this change would also affect many other structures as the algorithm is applied in an automated fashion. Theoretically, a fully adaptive morphological filter would account for heterogeneity within structures but is particularly challenging in 3D and another focus of our future work.

To depict volume changes in each label across the ADMBA, we plotted the change in volume from the mirrored to the smoothed (edge-aware) atlases by total volume for each label (Figure 4—figure supplement 2). By adapting the morphological filter sizes during both the edge-aware and final smoothing steps, we avoid label loss and minimize changes relative to label size as seen in the smaller labels (3,132 labels are present where x < 0.2). A caveat to the label preservation is that thin structures attached to larger regions can be lost as described above. We specifically designed the skeletonization step to address this issue to maximize retention of the original footprint including thin sections during the erosion process, from which to regrow all sections.

6) The accuracy of non-rigid registration of light-sheet images to the references is assessed only using a DSC value for whole-brain overlaps. This does not assess the precision of registration within the brain. The authors should apply some other measure to assess quality of alignment within the brain (e.g. mark internal landmarks visible in the reference and original light-sheet images, and measure the post-registration distance between them).

We note that since we have not changed the intensity images, other than to mirror them for symmetry, and our image registration is between the sample and atlas intensity images, our approach does not impact the image registration. We have assessed internal alignment through image registration by marking visible landmarks in the mirrored E18.5 atlas and each of the wild-type sample brains followed by measurement of the post-registration distances. Median distance across landmarks is 1,481 µm before and 135 µm after image registration (Author response image 3). The intensity coefficient of variation metric assesses the quality of the labels, rather than the intensity images or registration, although we expect that further improvements in registration would better demonstrate improvements in the atlases. It seems clear that there is substantial scope for improving registration to 3D images. To this end, we have also wondered whether application of the edge-aware approach post-registration could augment the registration process, though this would require a substantial rewrite of such algorithms.

**Author response image 3. sa2fig3:** Post-registration landmark distance for wild-type lightsheet images. Mean distances between manually annotated atlas and wild-type brain image landmarks are plotted for each brain after registration.